# AN INFORMATION-THEORETIC FRAMEWORK FOR OPTIMIZING EXPERIMENTAL DESIGN TO DISTINGUISH PROBABILISTIC NEURAL CODES

**Po-Chen Kuo & Edgar Y. Walker**
Department of Neurobiology and Biophysics
University of Washington
Seattle, WA 98195, USA
`{pckuo, eywalker}@uw.edu`

## ABSTRACT

The Bayesian brain hypothesis has been a leading theory in understanding perceptual decision-making under uncertainty. While extensive psychophysical evidence supports the notion of the brain performing Bayesian computations, how uncertainty information is encoded in sensory neural populations remains elusive. Specifically, two competing hypotheses propose that early sensory populations encode either the likelihood function (exemplified by probabilistic population codes) or the posterior distribution (exemplified by neural sampling codes) over the stimulus, with the key distinction lying in whether stimulus priors would modulate the neural responses. However, experimentally differentiating these two hypotheses has remained challenging, as it is unclear what task design would effectively distinguish the two. In this work, we present an information-theoretic framework for optimizing the task stimulus distribution that would maximally differentiate competing probabilistic neural codes. To quantify how distinguishable the two probabilistic coding hypotheses are under a given task design, we derive the *information gap*—the expected performance difference when likelihood versus posterior decoders are applied to neural populations—by evaluating the Kullback–Leibler divergence between the true posterior and a task-marginalized surrogate posterior. Through extensive simulations, we demonstrate that the information gap accurately predicts decoder performance differences across diverse task settings. Critically, maximizing the information gap yields stimulus distributions that optimally differentiate likelihood and posterior coding hypotheses. Our framework enables principled, theory-driven experimental designs with maximal discriminative power to differentiate probabilistic neural codes, advancing our understanding of how neural populations represent and process sensory uncertainty.

## 1 INTRODUCTION AND RELATED WORK

Effective perceptual decision-making requires organisms to process sensory information while accounting for the uncertainty inherent in the noisy and ambiguous sensory observations. The Bayesian brain hypothesis—with theoretical roots tracing to Laplace and von Helmholtz (de Laplace, 1820; Helmholtz, 1891)—proposes that the brain maintains internal generative models of the world and performs inference by computing probability distributions over task-relevant latent world states (Knill & Richards, 1996; Knill & Pouget, 2004). This framework has proven successful in explaining various aspects of human and animal perception, from multisensory integration and object recognition to motion perception and sensorimotor learning (Ernst & Banks, 2002; Weiss et al., 2002; Kersten et al., 2004; Alais & Burr, 2004; Körding & Wolpert, 2004). Extensive behavioral evidence demonstrates that humans and animals perform near optimally in perceptual tasks that require uncertainty estimation, strongly suggesting that sensory neural populations encode both task-relevant stimulus features and their associated uncertainty (Fiser et al., 2010; Pouget et al., 2013; Qamar et al., 2013; Ma & Jazayeri, 2014). However, the neural implementation of probabilistic computation remains actively debated, and how probability distributions are encoded and

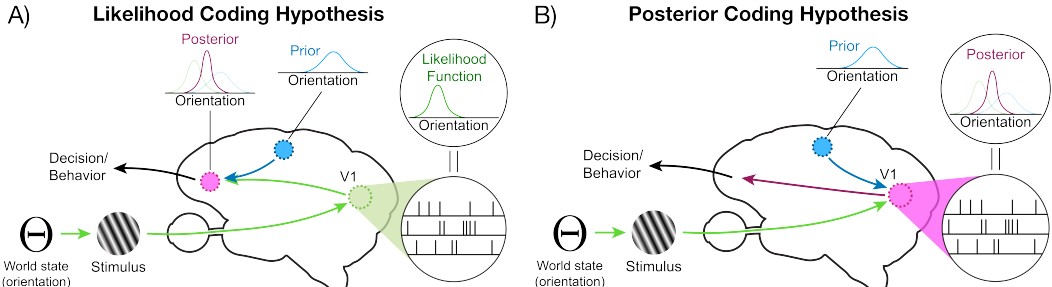

Figure 1: **Two competing hypotheses on how sensory uncertainty information is encoded in early sensory neural populations.** A) Likelihood coding hypothesis, exemplified by the probabilistic population code (Ma et al., 2006), proposes that early sensory populations encode the likelihood function over the stimulus, with posterior computation deferred to downstream areas. B) Posterior coding hypothesis, exemplified by the neural sampling code (Hoyer & Hyvärinen, 2002), posits that early sensory populations readily encode the posterior distribution over hidden world state by integrating prior knowledge conveyed via feedback connections from higher cortical areas.

represented in the brain is an area of active research (Yang & Shadlen, 2007; Orbán et al., 2016; Walker et al., 2020; Aitchison et al., 2021; Haefner et al., 2024).

An unresolved question concerns the format of probabilistic representations in sensory processing: Do early sensory populations encode the likelihood function over stimuli, or do they readily represent the posterior distribution that incorporate prior knowledge? The **likelihood coding hypothesis** (Fig. 1A) proposes that early sensory populations responding to stimuli (e.g., a drifting grating $x$) with underlying latent world states (e.g., orientation $\theta$) represent likelihood functions $L(\theta) \equiv p(x|\theta)$ (Jazayeri & Movshon, 2006; Walker et al., 2020). The classic form of probabilistic population code (Ma et al., 2006) exemplifies this hypothesis, proposing that sensory areas such as the primary visual cortex (V1) represent likelihood functions, accounting for the inherent variability in neural population responses. Previous work has demonstrated that likelihood functions decoded from V1 population responses are predictive of animals' trial-by-trial choices and reflects uncertainty associated with the sensory stimuli (Beck et al., 2008; Walker et al., 2020).

In contrast, motivated in part by the presence of extensive feedback connection from higher cortical areas that could convey existing 'prior' information, the **posterior coding hypothesis** (Fig. 1B) posits that sensory populations readily represent posterior distributions over latent world states $p(\theta|x)$, suggesting that even early sensory areas would incorporate the knowledge of priors to compute posterior distributions (Berkes et al., 2011; Festa et al., 2021). The neural sampling code (Hoyer & Hyvärinen, 2002) is one illustrative example in this category where a neural population is hypothesized to represent a posterior distribution by drawing a "sample" from the distribution and encoding it in its stochastic population responses, suggesting that neural variability naturally reflects the sampling process from a posterior distribution (Orbán et al., 2016; Haefner et al., 2016; Lange & Haefner, 2022; Shrinivasan et al., 2023).

The critical distinction between the two probabilistic coding hypotheses lies in whether stimulus priors $p(\theta)$ would modulate early sensory population responses. While existing studies have demonstrated that specific instantiations of each hypothesis can capture some aspects of observed neural response patterns (Haefner et al., 2016; Shivkumar et al., 2018; Walker et al., 2020), there is yet to be a targeted experiment aimed to directly distinguish the predictions from each coding hypothesis (Haefner et al., 2024). A fundamental challenge lies in identifying experimental designs—specifically, stimulus prior distributions—that would maximally differentiate the two coding hypotheses (Ma & Jazayeri, 2014). Since both coding hypotheses can often account for similar neural response patterns under traditional experimental conditions, targeted task designs where their predictions diverge maximally are crucial for distinguishing between likelihood and posterior coding hypotheses (Grabska-Barwinska et al., 2013; Shivkumar et al., 2018; Lange et al., 2023).

Motivated by research on optimal stimulus design for psychophysics studies (Watson & Pelli, 1983; Madigan & Williams, 1987), electrophysiology experiment (Lewi et al., 2006; 2011), and efficient

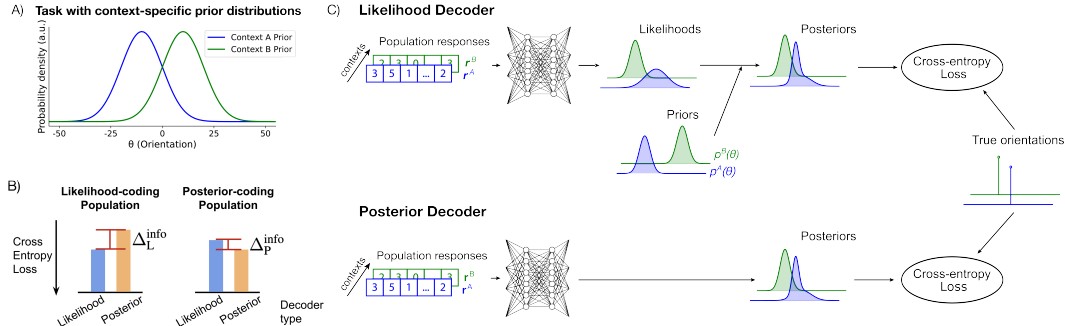

Figure 2: **A decoding approach to differentiating probabilistic neural codes.** A) An experimental paradigm consists of two contexts $c \in \{A, B\}$ with context-specific prior distributions $p^c(\theta)$. B) Information gap $\Delta^{\text{info}}$, the difference in likelihood (blue) and posterior (orange) decoder performances, can indicate whether the underlying neural population encodes the likelihood function (left) or the posterior distributions (right). C) Deep neural network-based decoders are used for decoding the likelihood function (top) or the posterior distribution (bottom) from population responses.

coding (Machens et al., 2005), in this work, we present an information-theoretic framework for designing experiments that optimally differentiate likelihood and posterior coding hypotheses. Our approach quantifies the expected difference in decodable information—which we term the **information gap** $\Delta^{\text{info}}$—when applying neural network-based decoders to extract likelihood or posterior information from sensory neural populations following either coding hypothesis. Specifically, we (1) derive analytic expressions for the information gap under each coding hypothesis, evaluated as the Kullback–Leibler (KL) divergence between the true posterior and a surrogate posterior utilizing Bayes-optimal estimators (Raventós et al., 2023); (2) validate theoretical predictions through simulations with deep neural network decoders applied to synthetic populations; (3) demonstrate how maximizing the information gap yields stimulus distributions that optimally differentiate the two probabilistic coding hypotheses; and (4) analyze existing neurophysiology datasets to show that conventional single-context experimental design is incapable of adjudicating the two hypotheses, supporting the necessity of the proposed optimized experimental framework.

Our framework provides a principled metric for optimizing experimental designs, establishing the theoretical upper bound on distinguishability between the two probabilistic coding hypotheses for a given task design. By maximizing this metric, we identify stimulus distributions that yield maximally differential decoder performance—providing rigorous, empirically testable predictions that directly adjudicate between competing theories of probabilistic neural representations.

## 2 INFORMATION GAP

We propose to determine whether early sensory populations encode likelihood functions or posterior distributions by examining how varying stimulus priors affects population responses. Classic orientation discrimination tasks under different contexts naturally involve altered stimulus prior distributions, making them ideal for testing this distinction (Qamar et al., 2013; Walker et al., 2020). Our experimental paradigm manipulates stimulus prior distributions across two different contexts and examines whether population responses vary according to changes in stimulus statistics (Fig. 2A)—a design that would leave likelihood-coding population responses invariant to an identical stimulus across contexts while systematically affecting posterior-coding population responses.

A decoding framework is leveraged to distinguish the probabilistic information content encoded in neural populations. As schematized in Fig. 2B, decoder performance degrades (increase in cross-entropy loss) when attempting to extract mismatched probabilistic content: if a neural population encodes likelihood functions, a decoder trained to extract likelihood information should outperform one extracting posterior information, and vice versa for posterior-coding populations. This differential performance between likelihood and posterior decoders thus serves as a diagnostic tool for identifying the underlying probabilistic representation. Building on recent advances in neural decoding (Walker et al., 2020), we employ deep neural network-based decoders that can effectively

extract the encoded information while incorporating the structural assumptions of each probabilistic coding hypothesis (Fig. 2D).

However, it is unclear what stimulus prior distributions would lead to maximal differentiabiltiy under the two probabilistic coding hypotheses (Fig. 8). While intuition suggests using maximally different context priors, this would limit stimulus overlap across contexts and thus prevent observing how different context priors modulate neural population responses to identical stimuli. This tradeoff—requiring sufficient prior differences to generate distinguishable population responses while maintaining adequate overlap for meaningful comparisons across contexts—cannot be resolved through intuition alone. To address this, we developed an information-theoretic framework that quantifies the expected decoder performance difference to systematically optimize experimental designs.

**Experimental paradigm** Consider a generative model of sensory observations $\theta \rightarrow x$, where $x$ represents noisy sensory observations (e.g. drifting gratings) generated according to the conditional distribution $p(x|\theta)$ given the hidden world state $\theta$ (e.g. orientation). We consider an experimental paradigm as introduced in Fig. 2A with two contexts $c = \{A, B\}$, each with its associated context frequency $p(c)$ and context-specific prior over the world state $p(\theta|c) \equiv p^c(\theta)$. We assume that the contexts of the current session are explicitly cued to ensure that subjects adopt the intended context-specific prior without having to infer the context from the stimuli. Once the subjects are trained on both contexts and the task performance stabilizes, we probe how early sensory populations represent probabilistic information.

Given neural population response vectors $r$, our goal is to assess the difference in decoder performances between a likelihood decoder $g_{\mathrm{L}}(r)$ and a posterior decoder $g_{\mathrm{P}}(r)$, optimized through minimizing the cross-entropy loss to extract likelihood functions and posterior distributions, respectively. We adopt an information-theoretic approach to derive the *expected* cross-entropy difference in decoder performance—a quantity we termed *information gap* $\Delta^{\mathrm{info}}$—for the two coding hypotheses under the theoretical limit of optimal decoders of probabilistic information. This quantity thus measures the expected increase in cross-entropy loss incurred when a decoder is forced to extract probabilistic content that is not actually encoded by the population responses. Although any empirical decoder would underestimate the true sensory information content, we posit that the derived theoretical limits would serve as reference points in evaluating the effectiveness of a task design in differentiating probabilistic neural codes. Below, we derive the information gap under each of the two probabilistic coding hypotheses.

**Information gap for likelihood coding hypothesis $\Delta_{\mathrm{L}}^{\mathrm{info}}$** Given discretized sensory observations $x \in \{x_i\}$, a task design specified by $(p(c), p^c(\theta)) \; \forall c \in \{A, B\}$, and a generative model $p(x_i|\theta)$, the expected difference in cross-entropy loss between optimal likelihood and posterior decoders, or the *information gap* $\Delta_{\mathrm{L}}^{\mathrm{info}}$ for a likelihood-coding population $r_{\mathrm{L}} \sim p(x|\theta)$, is derived as (see Appendix A.1 for full derivation):

$$
\begin{aligned}
\Delta_{\mathrm{L}}^{\mathrm{info}} &:= \mathbb{E}_{p(x_i,c)} \big[ D_{\mathrm{KL}}(p^c(\theta|x_i) \,||\, q_{P,i}^*(\theta)) \big] \\
&= \sum_{x_i} \Big\{ D_{\mathrm{KL}}(p^A(\theta|x_i) \,||\, q_{P,i}^*(\theta)) \cdot p(c=A) \Big[ \sum_{\theta} p(x_i|\theta) p^A(\theta) \Big] + \\
&\qquad D_{\mathrm{KL}}(p^B(\theta|x_i) \,||\, q_{P,i}^*(\theta)) \cdot p(c=B) \Big[ \sum_{\theta} p(x_i|\theta) p^B(\theta) \Big] \Big\}
\end{aligned} \tag{1}
$$

where $p^c(\theta|x_i)$ is the true posterior given observation $x_i$, which is the output of an optimal likelihood decoder. The surrogate posterior $q_{P,i}^*(\theta)$, which is the output of an optimal posterior decoder on likelihood-coding populations, is given by:

$$
q_{P,i}^*(\theta) = \frac{[p(c=A)p^A(\theta) + p(c=B)p^B(\theta)] \cdot p(x_i|\theta)}{\sum_{\theta'} \{[p(c=A)p^A(\theta') + p(c=B)p^B(\theta')] \cdot p(x_i|\theta')\}} \tag{2}
$$

Since likelihood-coding populations $r_{\mathrm{L}}$ contain no prior information, an optimal posterior decoder trained on such population cannot perfectly decode the posterior distribution. Instead, output of the optimal posterior decoder converges to a Bayes-optimal estimator as determined by marginalization over context distributions $p(c)$ and $p^c(\theta)$.

**Information gap for posterior coding hypothesis $\Delta_P^{\mathbf{info}}$**   Given discretized sensory observations $x \in \{x_i\}$, a task design specified by $(p(c), p^c(\theta)) \; \forall c \in \{A, B\}$, and a generative model $p(x_i|\theta)$, the expected difference in cross-entropy loss between optimal likelihood and posterior decoders, or *the information gap* $\Delta_P^{\mathrm{info}}$ for a posterior-coding population $\boldsymbol{r}_P \sim p(\theta|x)$, is derived as (see Appendix A.1 for full derivation):

$$
\begin{aligned}
\Delta_P^{\mathrm{info}} &:= \mathbb{E}_{p(x_i,c)}\big[D_{\mathrm{KL}}(p^c(\theta|x_i) \; || \; q_{L,i}^{c*}(\theta))\big] \\
&= \sum_{(x_j,x_k)} \Big\{ D_{\mathrm{KL}}(p^A(\theta|x_j) \; || \; q_{L,j}^{A*}(\theta)) \cdot p(c=A)\Big[\sum_\theta p(x_j|\theta)p^A(\theta)\Big] \\
&\qquad\qquad + D_{\mathrm{KL}}(p^B(\theta|x_k) \; || \; q_{L,k}^{B*}(\theta)) \cdot p(c=B)\Big[\sum_\theta p(x_k|\theta)p^B(\theta)\Big] \Big\}
\end{aligned}
\tag{3}
$$

where $p^c(\theta|x_i)$ is the true posterior given observation $x_i$, which is the output of an optimal posterior decoder. $q_{L,i}^{c*}(\theta)$ denotes a surrogate posterior which is the posterior distribution associated with the output of an optimal likelihood decoder. The sum in Eq. 3 includes only pairs $(x_j, x_k)$ that satisfy the condition expressed below in Eq. 4 as they are the only observations that would yield non-zero decoder performance difference. These are scenarios where identical population responses $\boldsymbol{r}_P$ (encoding the same posterior across the two contexts $c \in \{A, B\}$, i.e., $\boldsymbol{r}_{P,j}^A \approx \boldsymbol{r}_{P,k}^B$) must map to different likelihood functions ($p(x_j|\theta)$ and $p(x_k|\theta)$, respectively), preventing the optimal likelihood decoder from achieving perfect decoding. Observation pairs that do not satisfy Eq. 4 thus has no contribution to the sum in Eq. 3. The condition is given as:

$$
\forall_\theta, \; p^A(\theta|x_j) = p^B(\theta|x_k) \;\Leftrightarrow\; \forall_\theta, \; p^A(\theta) \cdot p(x_j|\theta) \propto p^B(\theta) \cdot p(x_k|\theta)
\tag{4}
$$

With this, the surrogate posterior distributions for the pair $(x_j, x_k)$ are given by:

$$
q_{L,j}^{A*}(\theta) = \frac{\ell_{jk}^*(\theta)p^A(\theta)}{Z_j^A[\ell_{jk}^*(\theta)]}, \quad q_{L,k}^{B*}(\theta) = \frac{\ell_{jk}^*(\theta)p^B(\theta)}{Z_k^B[\ell_{jk}^*(\theta)]}
$$

where $\ell_{jk}^*(\theta)$ denotes the output of the optimal likelihood decoder on the posterior-coding population, approaching a task-marginalized, Bayes-optimal estimator of the likelihood functions given by Eq. 5 below. $Z_j^A[\ell_{jk}^*(\theta)]$ and $Z_k^B[\ell_{jk}^*(\theta)]$ are normalization constants dependent on $\ell_{jk}^*(\theta)$, defined as $Z_j^A[\ell_{jk}^*(\theta)] := \sum_\theta p^A(\theta)\ell_{jk}^*(\theta)$ and $Z_k^B[\ell_{jk}^*(\theta)] := \sum_\theta p^B(\theta)\ell_{jk}^*(\theta)$.

The Bayes-optimal likelihood function estimator $\ell_{jk}^*(\theta)$ can be determined (up to a multiplicative constant) by solving the following implicit equation using fixed-point iteration (see A.1 for detail):

$$
\ell_{jk}^*(\theta) \propto \frac{\rho_j^A p^A(\theta|x_j) + \rho_k^B p^B(\theta|x_k)}{\frac{\rho_j^A}{Z_j^A[\ell_{jk}^*(\theta)]}p^A(\theta) + \frac{\rho_k^B}{Z_k^B[\ell_{jk}^*(\theta)]}p^B(\theta)}
\tag{5}
$$

where $\rho_j^A$ and $\rho_k^B$ denote the frequencies of each context conditioned on observed neural population responses of $\boldsymbol{r}_{P,j}^A$ or $\boldsymbol{r}_{P,k}^B$. Let us first define $S_j^A := p(c=A)\sum_\theta p^A(\theta)p(x_j|\theta)$ and $S_k^B := p(c=B)\sum_\theta p^B(\theta)p(x_k|\theta)$. Then the context frequencies $\rho_j^A$ and $\rho_k^B$ are given by:

$$
\rho_j^A := p(c=A|\boldsymbol{r} = \boldsymbol{r}_{P,j}^A \vee \boldsymbol{r}_{P,k}^B) = S_j^A/(S_j^A + S_k^B), \; \rho_k^B := p(c=B|\boldsymbol{r} = \boldsymbol{r}_{P,j}^A \vee \boldsymbol{r}_{P,k}^B) = S_k^B/(S_j^A + S_k^B)
$$

In summary, our information-theoretic framework quantifies the differentiability of the two probabilistic coding hypotheses under a given task design by deriving analytic expressions of the information gap $\Delta^{\mathrm{info}}$—the expected difference in decoder cross-entropy performances—for both likelihood coding hypothesis (Eq. 1) and posterior coding hypothesis (Eq. 3). The key insight stems from identifying the formula for the task-marginalized, Bayes-optimal estimators when decoding mismatched probabilistic information—that is, when decoding the posterior from a likelihood-coding population (Eq. 2) or when decoding the likelihood function from a posterior-coding population (Eq. 5).

Below, we empirically validate that information gaps accurately predict decoder performance differences under given task designs on simulated likelihood or posterior encoding neural populations across diverse task settings. We then demonstrate how maximizing the information gap enables targeted experimental designs that optimally differentiate the two probabilistic coding hypotheses.

## 3 SIMULATION EXPERIMENTS

To validate that the information gap accurately predicts decoder performance differences under both probabilistic coding hypotheses, we conducted comprehensive simulation experiments. We constructed synthetic likelihood-coding and posterior-coding neural populations, and applied likelihood and posterior decoders on these synthetic populations. These simulations serve two complementary purposes: validating our theoretical framework and providing practical insights into the scaling and convergence behavior of the information gap measure.

**Task design: Gaussian context priors**   We consider Gaussian context priors motivated by classic orientation-based discrimination experiments (Orbán et al., 2016; Walker et al., 2020). In this paradigm, subjects perform an orientation discrimination task with two contexts $c \in \{A, B\}$, with the context for each session sampled randomly, i.e. $p(c = A) = p(c = B) = 0.5$. Within each session, the trial-to-trial hidden world state $\theta$ (i.e. orientation) is drawn from context-specific Gaussian prior distributions $p^c(\theta) = \mathcal{N}(\mu^c, (\sigma^c)^2)$, where $\mu^c$ and $(\sigma^c)^2$ are task-specific parameters. In the simulation, we use identical variances for the two Gaussian priors $\sigma^A = \sigma^B = \sigma$.

We simulate noisy sensory observations $x$ by drawing from the conditional distribution defined by a given generative model $p(x|\theta)$. This stochastic process can be seen as capturing both intrinsic neuronal noise and extrinsic uncertainty in stimulus features. This generative model can be experimentally manipulated by varying stimulus parameters such as contrast. Indeed, lower contrast induces increase in observation variance, reflecting increased sensory uncertainty. In the simulation, $p(x|\theta)$ is modeled as Gaussian distributions to reflect Gaussian-like orientation tuning curves commonly observed in V1 neurons and to capture the effect of different contrast levels by systematically varying the standard deviation (Walker et al., 2020).

For simulated population responses, we implemented Poisson neuron models with Gaussian tuning curves (Walker et al., 2020). Likelihood-coding population's mean firing rates $\boldsymbol{r}_{\mathrm{L}}$ are encoded through Gaussian tuning curves based on the sampled sensory observations $x$, while posterior-coding population's mean firing rates $\boldsymbol{r}_{\mathrm{P}}$ are additionally modulated by the context-specific prior $p^c(\theta)$, effectively encoding the posterior $p^c(\theta|x) \propto p(x|\theta) \cdot p^c(\theta)$. In both cases, spike counts were then generated by sampling from Poisson distribution with the given mean firing rate. We additionally considered a more complex, bio-realistic gain-modulated Poisson neuron model for simulating population responses Goris et al. (2014). As shown in Fig. 2C, deep neural networks are trained with cross-entropy loss to serve as flexible, powerful decoders of probabilistic distributions, decoding either the likelihood function or the posterior from the simulated neural population responses (Walker et al., 2020). See A.3 for full details of the simulation experiments and decoder setups.

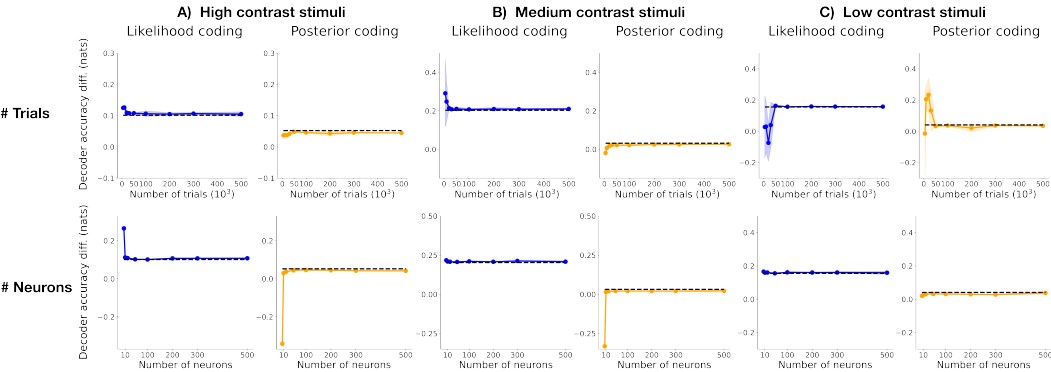

Figure 3: **Decoder performance difference on simulated populations converges to the theoretical prediction of information gap.** A) On simulated neural populations encoding the likelihood function (left, blue) or the posterior distributions (right, orange) responding to high contrast stimuli, the difference between the likelihood and posterior decoder performances converges to the theoretical value of information gap (dashed lines) as the total number of trials increases (top, with fixed number of neurons = 500), and as the total number of neurons in the population increases (bottom, with fixed number of trials = 30k). (shaded areas denote the s.t.d. across 5 random seeds.) B) Same for medium contrast stimuli and C) for low contrast stimuli.

**Scaling and convergence** We first examine the scaling and convergence properties of the theoretical prediction of information gap (see additional analyses and ablation studies in A.6 and A.7). Fig. 3 demonstrates convergence of the empirical decoder performance differences on simulated Poisson neural populations across various stimulus contrast levels. For a given set of task parameters, decoder performance differences for both simulated populations—likelihood-coding (blue) and posterior-coding (orange) populations—rapidly converge to the theoretically derived information gap (dashed lines) computed via Eq. 1 and 3, as the number of trials increases (top) and as the number of neurons increases (bottom). This empirical convergence suggests that the information gap measure derived from our framework accurately predicts the asymptotic decoder performance difference quantifying the effectiveness of a task design.

**Validation across parameter space** We next assess the validity of the theoretical prediction of information gap across a wide range of simulation settings. Across different levels of stimulus contrast, at least ten different sets of task parameters are selected to compute the theoretical value of information gap and to simulate likelihood and posterior encoding populations. Fig. 4 systematically compares the theoretical predictions of information gap and the empirical decoder performance difference across diverse task design parameters, both under the Poisson neural model (top) and under the more complex, gain-modulated Poisson neural model (bottom, Goris et al. (2014)). On both types of simulated neural models and across different contrast levels, the comparison reveals remarkable agreement between the information gap prediction and the empirical decoder performance difference for both likelihood and posterior coding hypotheses.

Notably, information gaps for likelihood-coding populations ($\Delta_L^{\text{info}}$) exceed those for posterior-coding populations ($\Delta_P^{\text{info}}$) by up to an order of magnitude. Our framework provides an intuitive explanation: for likelihood coding hypothesis, every observation contributes to the information gap calculation, whereas for posterior coding hypothesis, only pairs satisfying Eq. 4 contribute to the estimate. This asymmetry suggests that distinguishing posterior-coding populations presents greater experimental challenges, requiring careful task design to achieve sufficient statistical power.

Overall, these simulation results establish that our information-theoretic framework accurately predicts decoder performance differences for neural populations following either probabilistic coding hypothesis, providing a quantitative foundation for designing targeted, theory-driven experiments to differentiate probabilistic neural representations in early sensory areas.

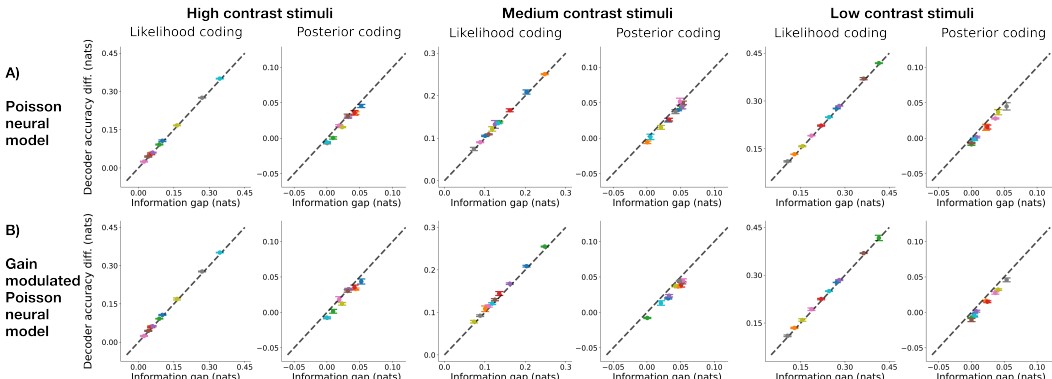

Figure 4: **Information gap accurately predicts decoder performance difference on simulated populations across diverse task settings.** A) On simulated Poisson neural populations responding to high (left), medium (middle), and low (right) contrast stimuli, theoretical values of information gap (x-axis) accurately predicts the decoder performance difference on simulated neural populations (y-axis) across multiple task design parameters, for both the likelihood-coding and posterior-coding populations. (Each color marks one set of task parameters used for both types of simulated populations; Error bars denote the s.t.d. across 5 random seeds.) B) Same for simulated populations using a more complex, bio-realistic gain-modulated Poisson neural model (Goris et al., 2014).

# 4 TASK OPTIMIZATION TO DIFFERENTIATE PROBABILISTIC NEURAL CODES

Given the strong agreement between the empirical decoder performance differences and the theoretical information gap measure, we now demonstrate how to optimize task designs to maximally differentiate the two probabilistic coding hypotheses. The goal is to systematically explore the task parameter space to identify task parameters that would yield maximum information gap.

## 4.1 INFORMATION GAP LANDSCAPE FOR GAUSSIAN CONTEXT PRIORS

For tasks with Gaussian context priors, we evaluate the information gap across the two-dimensional task parameter space defined by (1) the distance between the two Gaussian means $d = |\mu^A - \mu^B|$, and (2) the shared standard deviation for both Gaussian priors $\sigma$ (Fig. 8). The landscapes of information gap across three different contrast levels are shown in Fig. 5, for both likelihood-coding populations (top) and posterior-coding populations (bottom). We first observe that the information gap landscape depends on the stimulus contrast level, suggesting that experimental design should be tailored to specific stimulus features such as contrast. In addition, decreasing contrast expands the parameter region yielding substantial information gaps for both probabilistic codes, agreeing with the intuition that prior information becomes more influential when sensory observations alone provide insufficient information for reliable inference.

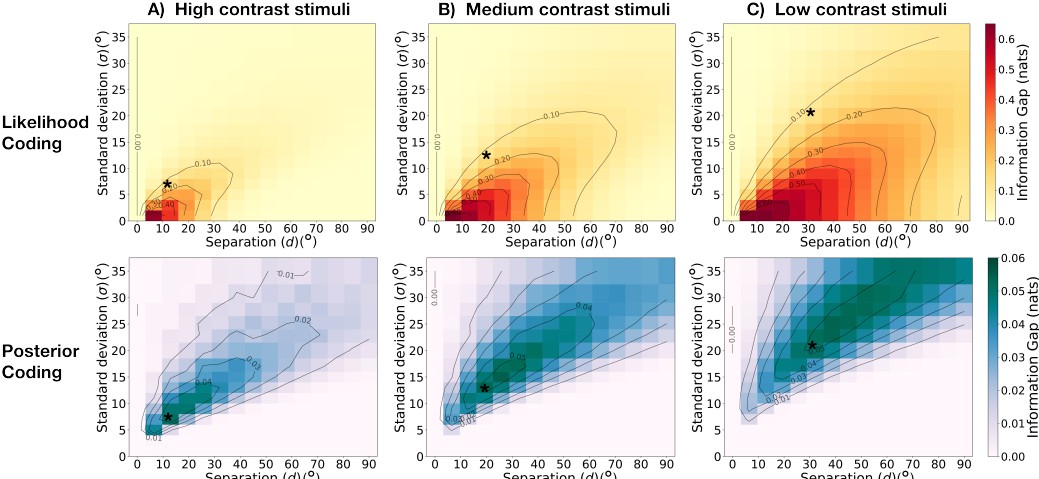

Figure 5: **Information gap landscapes inform practical task designs that optimally differentiate probabilistic representations in neural populations.** A) Information gap as a function of task parameters ($d$: separation between context priors, and $\sigma$: context prior standard deviations) for both the likelihood coding hypothesis (top) and the posterior coding hypothesis (bottom) when presented with high contrast stimuli. The asterisks identify strategic task designs that achieve the tradeoff where posterior-coding information gap approaches its maximum while likelihood-coding maintains sufficient discriminative signal. B) Same for medium contrast stimuli and C) for low contrast stimuli.

**Strategic task design** Crucially, for a given contrast level, the information gap landscape depends on the underlying probabilistic coding hypothesis, revealing a trade-off when optimizing experimental design: task parameters that maximize the discriminability for likelihood-coding populations diverge from those optimal for posterior-coding populations. This divergence necessitates a strategic selection of parameters that balance discriminative power across both hypotheses. Considering the notable asymmetry in information gap magnitudes—with posterior-coding values typically an order of magnitude smaller than likelihood-coding ones, one might prioritize parameters that maximize posterior-coding discriminability while maintaining adequate likelihood-coding sensitivity. The asterisks in Fig. 5 identify such strategic "sweet spots" where posterior-coding information gap $\Delta_P^{info}$ approaches its maximum while likelihood-coding information gap $\Delta_L^{info}$ maintains sufficient discriminative signal. For low contrast stimuli, such optimization occurs with prior separation of

$d \approx 30°$ and standard deviation of $\sigma \approx 20°$. As contrast increases, the optimal task parameters shift toward smaller prior separations and narrower standard deviations.

## 4.2 Information gap landscape for non-Gaussian context priors

We next explore the feasibility of using other types of stimulus context prior distributions. Specifically, we test the effectiveness of heavy-tailed priors such as student's t-distribution and Cauchy distribution (see A.9 for analysis on thin-tailed priors). Fig. 6 shows the information gap landscape under medium contrast using the student's t-distribution (top) or the Cauchy distribution (bottom) as stimulus priors. Compared to Gaussian priors, areas with substantial information gap become more limited. In particular, posterior-coding information gap is zero almost throughout the entire parameter space, indicating that heavy-tailed priors are not suitable for distinguishing posterior-coding populations. Our theoretical framework provides an explanation: under heavy-tailed priors, there are barely any observation pairs satisfying Eq. 4 that would contribute to the information gap (See A.8 for details). Finally, almost no overlap exists between areas where $\Delta^{\text{info}}$ for each coding hypothesis is maximized, suggesting that any choice of task parameters optimal for identifying one hypothesis will necessarily sacrifice the effectiveness of identifying the other. Overall, this analysis suggests that heavy-tailed priors are not ideal for differentiating probabilistic coding hypotheses.

In summary, our framework transforms parameter selection from heuristic search to principled optimization, directly identifying task designs that maximize statistical power for differentiating probabilistic neural representations. The resulting information gap landscapes can guide targeted experiments toward parameter combinations most likely to yield decisive empirical results.

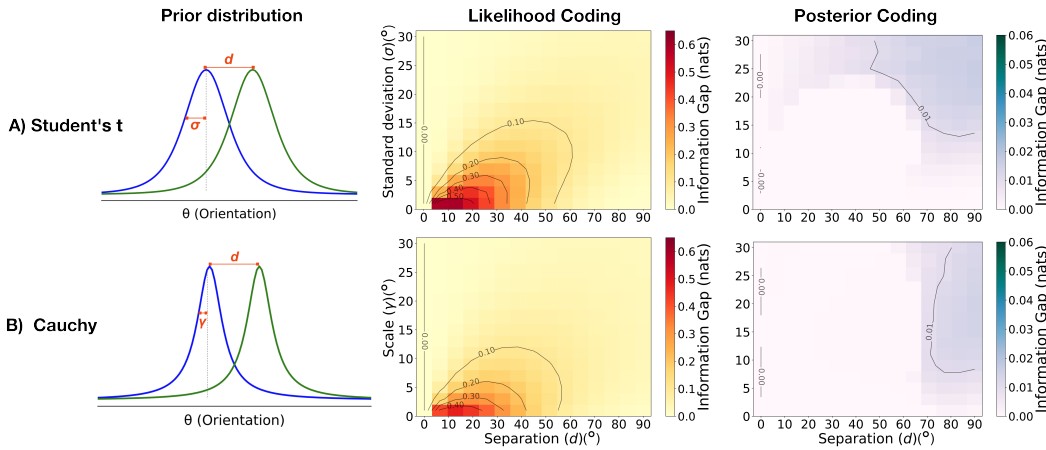

Figure 6: **Information gap landscape suggests heavy tailed distributions are not ideal stimulus prior distributions for differentiating coding hypotheses.** A) Using student's t-distribution with degrees of freedom $\nu = 3$ as stimulus priors (left), information gap under medium contrast stimuli as a function of task parameters (separation $d$ and standard deviations $\sigma$) for both the likelihood coding hypothesis (middle) and the posterior coding hypothesis (right) shows decreased information gap with minimal overlap compared to task design with Gaussian context priors. B) Same for Cauchy distribution as stimulus priors with task parameters separation $d$ and scale $\gamma$.

## 5 Empirical results on neurophysiology data

Distinguishing the two probabilistic coding hypotheses depends on how population responses change across contexts with different priors, yet existing datasets mostly provide only a single fixed stimulus context without manipulation in priors. To demonstrate that existing neurophysiology datasets with single-context experimental designs cannot adjudicate the two coding hypotheses, we report empirical results on orientation decoding from real neural data using the Allen Brain Observatory Visual Coding Neuropixels Dataset (Siegle et al., 2021). Under such single-context experimental design with uniform prior, our theory predicts no performance difference between the likelihood and posterior decoders, i.e. $\Delta^{\text{info}} = 0$. In Fig. 7, we performed orientation decoding

analysis on the Allen Visual Coding dataset. The result shows indistinguishable performance between the likelihood and posterior decoders (difference $= 0.0024 \pm 0.064$, $p = 0.63$), which agrees with our theoretical prediction. In fact, previous decoding work on macaque V1 similarly discussed why their experimental design resulted in ambiguity in differentiating coding hypotheses due to the lack of multiple context priors (Walker et al., 2020). This result on empirical data underscores why future experiments incorporating context-dependent prior manipulations will be essential for adjudicating the competing probabilistic coding hypotheses and how our proposed targeted experimental paradigm could help optimize the task design for such experiments.

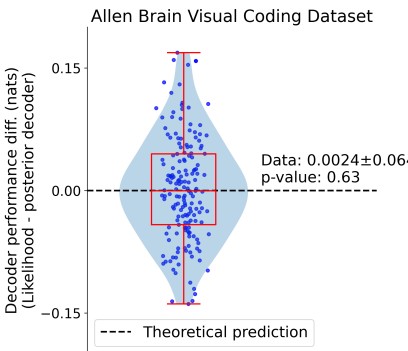

Figure 7: **Decoding analysis on the Allen Visual Coding datasets (Siegle et al., 2021) shows indistinguishable decoder performance difference.** Across 169 sessions with large enough trials ($> 300$ trials), the decoder cross-entropy performance difference (likelihood decoder - posterior decoder) is $0.0024 \pm 0.064$, which is not significantly different from the model prediction of $0$ ($p = 0.63$) (Each dot indicates one session.). This empirical result highlights the necessity of the context-dependent prior manipulation for distinguishing probabilistic coding hypotheses.

## 6 DISCUSSION AND CONCLUSIONS

We presented an information-theoretic framework for optimizing experimental design to address whether early sensory neural populations encode likelihood functions or posterior distributions. We derive analytic expressions for *information gap*—the expected decoder performance difference when extracting mismatched probabilistic content. This measure quantifies how effectively an experimental design can distinguish between competing probabilistic coding hypotheses, providing precise predictions validated through extensive simulations. Most critically, maximizing the information gap yields principled experimental designs that can optimally discriminate between probabilistic neural codes, enabling decisive experiments to resolve a fundamental debate about Bayesian computation in the brain. By developing theoretical framework to quantify how well experiments can distinguish between competing coding hypotheses, this approach demonstrates how computational theory can directly guide experimental neuroscience.

Although our approach follows the popular ideal observer framework for studying perceptual decision-making under uncertainty (Ma et al., 2006; Beck et al., 2008; Walker et al., 2020), it could naturally be extended to incorporate imperfect priors given empirical psychophysical results. The imperfect or biased prior of a subject can be inferred by analyzing their psychophysical curves, which can subsequently be used in place of the ground-truth prior to adjust the calculation of the information gap (See the detailed procedure in A.4 and Fig. 10). Furthermore, it should be noted that likelihood and posterior coding are not the only relevant theories to be considered, but they can be regarded as two extremes that differ in the probabilistic quantity encoded by the sensory population. Consequently, *mixed or intermediate* hypotheses that lie somewhere between the two canonical hypotheses may be proposed, where the neural population encodes some mixture of likelihood and posterior (for example, Ganguli & Simoncelli (2010)). In A.5 and Fig. 11, we discuss how our framework and optimized tasks can be extended to discriminate more nuanced, mixed coding hypotheses. By optimizing task parameters to maximally separate the canonical hypotheses, we simultaneously maximize sensitivity to discriminating more nuanced probabilistic coding theories.

**Scope and limitations** To compute information gap, our framework requires reasonable generative models and thus may require prior work establishing neural response properties. In addition, the decoding approach requires sufficient neural population response data for training. Our framework also provides a foundation that can be extended in several directions: 1) The framework extends beyond orientation-based stimuli to continuous observations and other types of distributions through numerical methods; and 2) Incorporating more bio-realistic neural models such as noise correlations and nonlinearities would further strengthen predictions.

**Reproducibility statement**  The source code for computing the information gap, implementing both likelihood and posterior decoders, and running all simulation experiments is available at https://github.com/walkerlab/information-gap-probabilistic-neural-codes. The detailed derivation of information gap is presented in A.1. The details of simulation experiments including synthetic neural population response generation procedures, deep neural network decoder architectures and hyper-parameters, and training procedures are presented in A.3.

**Acknowledgements**  We thank Patrick Zhang and Suhas Shrinivasan for their insightful feedback and discussions. We thank Daniel Sitonic for the technical support. We gratefully acknowledge the community support from the University of Washington Computational Neuroscience Center and the Allen Institute for Neural Dynamics. We also thank the anonymous reviewers for their valuable and constructive review.

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

# A  TECHNICAL APPENDICES AND SUPPLEMENTARY MATERIAL

## A.1  INFORMATION GAP DERIVATION

Consider a generative model of sensory observations $\theta \to x$, where $x$ is the noisy sensory observation (e.g. a drifting grating stimulus) generated according to the conditional distribution $p(x|\theta)$, where $\theta$ is the hidden state of the environment (e.g. true orientation of the drifting grating stimulus). Note that the likelihood function is given by $\mathcal{L}(\theta) \equiv p(x|\theta)$ for a specific observation $x$. Consider an experimental setup where there are two possible stimulus generation contexts: $c = \{A, B\}$ with their associated context-specific latent priors $p^c(\theta) := p(\theta|c)$ and their context frequencies $p(c)$.

Given a sensory observation $x$ and a context $c$, the context-dependent posterior distribution of $\theta$, denoted as $p^c(\theta|x) := p(\theta|x, c)$, is given by the Baye's rule:

$$
\begin{aligned}
p^c(\theta|x) &= \frac{p^c(\theta, x)}{p^c(x)} \\
&= \frac{p^c(x|\theta) \cdot p^c(\theta)}{\sum_{\theta'} p^c(x|\theta') \cdot p^c(\theta')}, \quad \text{Since the generative process } \theta \to x \text{ is independent of } c, \\
&= \frac{p(x|\theta) \cdot p^c(\theta)}{\sum_{\theta'} p(x|\theta') \cdot p^c(\theta')} \\
&\propto p(x|\theta) \cdot p^c(\theta)
\end{aligned}
$$

For a given neural population response vector $\boldsymbol{r}$, consider two competing probabilistic coding hypothesis:

1.  **Likelihood coding hypothesis**: $\boldsymbol{r}_{\mathrm{L}} \sim p(x|\theta)$, where the neural population responses $\boldsymbol{r}_{\mathrm{L}}$ is hypothesized to encode the likelihood function of the stimulus $p(x|\theta)$.

2.  **Posterior coding hypothesis**: $\boldsymbol{r}_{\mathrm{P}} \sim p(\theta|x)$, where the neural population response $\boldsymbol{r}_{\mathrm{P}}$ is hypothesized to encode the posterior distribution of the hidden state given the stimulus $p(\theta|x)$.

We consider whether it is possible to differentiate the probabilistic information content encoded in given neural population responses $\boldsymbol{r}$ through a decoding approach. Intuitively, if a neural population is encoding the likelihood function, then a decoder decoding the likelihood function should lead to a better performance then a decoder decoding the posterior distribution; vice versa if the neural population is encoding the posterior distribution. In other words, decoder performance degrades when trying to decode mismatched probabilistic content, such that the difference in decoder performance when decoding the likelihood function versus decoding the posterior distribution can be used to differentiate whether a given neural population is encoding the likelihood function (likelihood coding hypothesis) or the posterior distribution (posterior coding hypothesis). Below, we formalize this intuition by deriving the expected decoder performance difference.

Consider applying a decoder function $g$ which is optimized to decode some probabilistic information content from the neural population responses under cross-entropy loss:

$$
g(\boldsymbol{r}) \longrightarrow p(\cdot) \quad \text{where } g \text{ is a decoder function}
$$

Note that to establish the expected difference between decoder performances, we assume ideal decoders in derivations. Empirically, we assume the decoder is expressive enough (e.g. a multi-layer perceptron, MLP) and fully trained, and the data is abundant such that the performance of the decoder would closely approximate that of the ideal decoder.

Adopting an information-theoretic approach, our goal is to derive the *expected difference* between decoder performances as measured in cross-entropy when decoding the likelihood function versus decoding the posterior distribution from given neural population responses $\boldsymbol{r}$, a quantity that we termed the *information gap*, $\Delta^{\mathrm{info}}$, between the two decoders under a given experimental design specified by $(p(c), p^c(\theta)), \forall c \in \{A, B\}$ and a generative model $p(x|\theta)$. Below we will separately derive the information gap for likelihood coding hypothesis, $\Delta_{\mathrm{L}}^{\mathrm{info}}$, and the information gap for posterior coding hypothesis, $\Delta_{\mathrm{P}}^{\mathrm{info}}$, respectively. As a by-product, our information-theoretic analysis framework also allows for deriving the *expected decoder performance* for each decoder under the limit of perfect decoding as measured in cross-entropy.

### A.1.1 INFORMATION GAP FOR LIKELIHOOD CODING HYPOTHESIS $\Delta_{\mathrm{L}}^{\mathrm{INFO}}$

For a likelihood coding population, the neural population responses $r_{\mathrm{L}}$ encode the likelihood function of the sensory stimulus, which are not modulated by and hence independent of the context prior.

$$r_{\mathrm{L}} \sim f(p(x|\theta)), \text{ where } f \text{ is some neural encoding function.}$$

Note that since the decoders are optimized under cross-entropy loss:

$$H(p,q) = -\mathbb{E}_p[\log q] = H(p) + D_{\mathrm{KL}}(p \,||\, q)$$

when $q^* = p \Leftrightarrow H(p, q^*)$ is minimized.

**Decoding performance of a perfect likelihood decoder $g_L$**

Applying a likelihood decoder $g_L$ to a likelihood coding population $r_{\mathrm{L}}$, we want

$$g_L(r_{\mathrm{L}}) \longrightarrow p(x|\theta)$$

Let us assume the observation space can be discretized into $x \in \{x_i\}$, and consider the neural population responses associated with each $x_i$:

$$\forall x_i, c : r_{\mathrm{L},i}^c = r_{\mathrm{L},i} \sim f(p(x_i|\theta))$$

Since $r_{\mathrm{L},i}$ is context-independent, let us denote the likelihood decoder output $g_L(r_{\mathrm{L},i}^c) = g_L(r_{\mathrm{L},i})$. Since the ground truth context prior $p^c(\theta)$ is provided to the likelihood decoder $g_L$ as schematized in Fig. 2, with the likelihood decoder output $g_L(r_{\mathrm{L},i})$ and the corresponding context prior $p^c(\theta)$, the context-dependent decoded posterior distribution $q_{L,i}^c(\theta)$ is given by:

$$q_{L,i}^c(\theta) = \eta_{L,i}^c \cdot g_L(r_{\mathrm{L},i}) \cdot p^c(\theta), \text{ where } \eta_{L,i}^c \text{ is a normalization constant.}$$

The cross-entropy loss for data samples associated with $x_i, c$, i.e. $H(p^c(\theta|x_i), q_{L,i}^c(\theta))$, is minimized when:

$$q_{L,i}^{c*}(\theta) = p^c(\theta|x_i)$$

$$\Rightarrow \eta_{L,i}^c \cdot g_L^*(r_{\mathrm{L},i}) \cdot p^c(\theta) = \frac{p(x_i|\theta) \cdot p^c(\theta)}{p(x_i)}$$

$$\Rightarrow g_L^*(r_{\mathrm{L},i}) = \alpha_{L,i}^c \cdot p(x_i|\theta), \text{ where } \alpha_{L,i}^c \text{ is a constant} \tag{6}$$

That is, after training, the likelihood decoder output $g_L(r_{\mathrm{L},i})$ will converge to $g_L^*(r_{\mathrm{L},i}) \propto p(x_i|\theta)$ given enough samples.

To get the expected cross-entropy loss across the entire data set, we marginalizing over all $x_i, c$, and the expected cross-entropy loss for a perfect likelihood decoder can be evaluated as:

$$\begin{aligned}
\mathbb{E}_{p(x_i,c)}[H(p^c(\theta|x_i), q_{L,i}^{c*}(\theta))] &= \mathbb{E}_{p(x_i,c)}[H(p^c(\theta|x_i)) + D_{\mathrm{KL}}(p^c(\theta|x_i) \,||\, q_{L,i}^{c*}(\theta))] \\
&= \mathbb{E}_{p(x_i,c)}[H(p^c(\theta|x_i))] \\
&= \sum_{x_i,c} H(p^c(\theta|x_i)) \cdot p(x_i, c) \\
&= \sum_{x_i,c} H(p^c(\theta|x_i)) \cdot p(c) \Big[ \sum_\theta p(x_i|\theta) p^c(\theta) \Big] \\
&= \sum_{x_i} \sum_c H(p^c(\theta|x_i)) \cdot p(c) \Big[ \sum_\theta p(x_i|\theta) p^c(\theta) \Big] \\
&= \sum_{x_i} \Big\{ H(p^A(\theta|x_i)) \cdot p(c = A) \Big[ \sum_\theta p(x_i|\theta) p^A(\theta) \Big] + \\
&\qquad\qquad H(p^B(\theta|x_i)) \cdot p(c = B) \Big[ \sum_\theta p(x_i|\theta) p^B(\theta) \Big] \Big\}
\end{aligned} \tag{7}$$

where the second equality holds because $D_{\mathrm{KL}}(p^c(\theta|x_i) \,||\, q_{L,i}^{c*}(\theta)) = 0$ for a perfect likelihood decoder as derived above. That is, the expected cross-entropy loss for a perfect likelihood decoder should approach the expected posterior entropy as determined by the context frequencies and context prior distributions as given by Eq. 7.

**Decoding performance of the best possible posterior decoder** $g_P$

Applying a posterior decoder $g_P$ to a likelihood coding population $\boldsymbol{r}_{\mathrm{L}}$, we want:

$$g_P(\boldsymbol{r}_{\mathrm{L}}) \longrightarrow p^c(\theta|x)$$

However, since there is no context information encoded in the population responses $\boldsymbol{r}_{\mathrm{L}}$, the posterior decoder $g_P$ cannot achieve the same performance as the likelihood decoder in Eq. 7, as there are scenarios where identical inputs ($\boldsymbol{r}_{\mathrm{L},i}$) are trained to map to different outputs ($p^c(\theta|x_i)$) depending on the inaccessible ground-truth context information $p^c(\theta)$.

Let us consider the neural population responses associated with each observation $x_i$:

$$\forall x_i, c : \boldsymbol{r}_{\mathrm{L},i}^c = \boldsymbol{r}_{\mathrm{L},i} \sim f(p(x_i|\theta))$$

The frequency of a context given the observation of data samples associated with $x_i$ is given by:

$$
\begin{aligned}
p(c|x = x_i) &= \frac{p(c, x_i)}{p(x_i)} \\
&= \frac{p(c) \cdot p(x_i|c)}{\sum_{c'} p(c') \cdot p(x_i|c')} \\
&= \frac{p(c) \cdot \sum_\theta p^c(\theta) \cdot p(x_i|\theta)}{\sum_{c'} p(c') \sum_\theta p^{c'}(\theta) \cdot p(x_i|\theta)}
\end{aligned}
$$

Let us denote

$$S_i^A := p(c = A) \sum_\theta p^A(\theta) p(x_i|\theta)$$

$$S_i^B := p(c = B) \sum_\theta p^B(\theta) p(x_i|\theta)$$

Hence, we can define the observation-dependent context frequency for a given $x_i$ as:

$$\rho_i^A := p(c = A|x = x_i) = S_i^A/(S_i^A + S_i^B)$$
$$\rho_i^B := p(c = B|x = x_i) = S_i^B/(S_i^A + S_i^B)$$

Now, let us denote the posterior decoder output $q_{P,i}(\theta) := g_P(\boldsymbol{r}_{\mathrm{L},i})$, highlighting that the output can be interpreted directly as the posterior distribution over the hidden state $\theta$, as schematized in Fig. 2. Since the posterior decoder output is agnostic to the specific context and the associated prior $p^c(\theta)$, under cross-entropy loss, $q_{P,i}(\theta)$ is trained to minimize the expression below:

$$
\begin{aligned}
&\min_{q_{P,i}(\theta)} \left\{ \mathbb{E}_{p(c|x_i)} \left[ H(p^c(\theta|x_i), q_{P,i}(\theta)) \right] \right\} \\
=& \min_{q_{P,i}(\theta)} \left\{ \rho_i^A H(p^A(\theta|x_i), q_{P,i}(\theta)) + \rho_i^B H(p^B(\theta|x_i), q_{P,i}(\theta)) \right\} \\
=& \min_{q_{P,i}(\theta)} \left\{ -\sum_\theta \left[ \rho_i^A p^A(\theta|x_i) \cdot \log q_{P,i}(\theta) + \rho_i^B p^B(\theta|x_i) \cdot \log q_{P,i}(\theta) \right] \right\} \\
=& \min_{q_{P,i}(\theta)} \left\{ -\sum_\theta \left[ \rho_i^A p^A(\theta|x_i) + \rho_i^B p^B(\theta|x_i) \right] \cdot \log q_{P,i}(\theta) \right\}
\end{aligned}
$$

Since $p^A(\theta|x_i)$ and $p^B(\theta|x_i)$ are both probability distributions over $\theta$, and $\rho_i^A + \rho_i^B = 1$, the expression $\rho_i^A p^A(\theta|x_i) + \rho_i^B p^B(\theta|x_i)$ represents a proper probability distribution over $\theta$. Therefore the loss above is minimized when:

$$
\begin{aligned}
q_{P,i}^*(\theta) &= \rho_i^A p^A(\theta|x_i) + \rho_i^B p^B(\theta|x_i) \\
&= \frac{S_i^A}{S_i^A + S_i^B} \frac{p^A(\theta) p(x_i|\theta)}{\sum_\theta p^A(\theta) p(x_i|\theta)} + \frac{S_i^B}{S_i^A + S_i^B} \frac{p^B(\theta) p(x_i|\theta)}{\sum_\theta p^B(\theta) p(x_i|\theta)} \\
&= \frac{[p(c = A) p^A(\theta) + p(c = B) p^B(\theta)] \cdot p(x_i|\theta)}{S_i^A + S_i^B} \\
&= \frac{[p(c = A) p^A(\theta) + p(c = B) p^B(\theta)] \cdot p(x_i|\theta)}{\sum_{\theta'} \{ [p(c = A) p^A(\theta') + p(c = B) p^B(\theta')] \cdot p(x_i|\theta') \}}
\end{aligned}
$$

That is, after training, the best possible posterior decoder output for data samples associated with $x_i$, i.e. $q_{P,i}^*(\theta)$, is as if the decoder were to use a surrogate prior:

$$\tilde{p}_i(\theta) = p(c = A)p^A(\theta) + p(c = B)p^B(\theta) \tag{8}$$

which is the task-marginalized, Bayes-optimal estimator of the prior distributions over $\theta$ across contexts $c \in \{A, B\}$. Interestingly, this surrogate prior distribution is independent of $x_i$.

Since likelihood-coding populations $\boldsymbol{r}_L$ contain no prior information $p^c(\theta)$, a posterior decoder $g_P$ trained on such population responses cannot perfectly decode the posterior distribution. Instead, the posterior decoder output converges to a Bayes-optimal estimate of context-dependent posteriors determined by the context distributions $p(c)$ and $p^c(\theta)$. To obtain the expected cross-entropy loss across the entire data set, we marginalize over all $x_i, c$, yielding:

$$
\begin{aligned}
\mathbb{E}_{p(x_i,c)}[H(p^c(\theta|x_i), q_{P,i}^*(\theta))] &= \mathbb{E}_{p(x_i,c)}[H(p^c(\theta|x_i)) + D_{\mathrm{KL}}(p^c(\theta|x_i) \,\|\, q_{P,i}^*(\theta))] \\
&= \mathbb{E}_{p(x_i,c)}[H(p^c(\theta|x_i))] + \mathbb{E}_{p(x_i,c)}\big[D_{\mathrm{KL}}(p^c(\theta|x_i) \,\|\, q_{P,i}^*(\theta))\big] \\
&= \text{CE loss of the perfect likelihood decoder (Eq. 7)} \\
&\quad + \mathbb{E}_{p(x_i,c)}\big[D_{\mathrm{KL}}(p^c(\theta|x_i) \,\|\, q_{P,i}^*(\theta))\big]
\end{aligned} \tag{9}
$$

**Information gap for a likelihood coding population $\Delta_{\mathrm{L}}^{\mathbf{info}}$**

From Eq. 9, let us define $\Delta_{\mathrm{L}}^{\mathrm{info}}$, the information gap between a perfect likelihood decoder ($g_L^*$) and the best possible posterior decoder ($g_P^*$) applied on a likelihood-coding population, evaluated as the expected difference in cross-entropy loss between the two decoders:

$$
\begin{aligned}
\Delta_{\mathrm{L}}^{\mathrm{info}} &:= \mathbb{E}_{p(x_i,c)}\big[D_{\mathrm{KL}}(p^c(\theta|x_i) \,\|\, q_{P,i}^*(\theta))\big] \\
&= \sum_{x_i,c} D_{\mathrm{KL}}(p^c(\theta|x_i) \,\|\, q_{P,i}^*(\theta)) \cdot p(x_i, c) \\
&= \sum_{x_i,c} D_{\mathrm{KL}}(p^c(\theta|x_i) \,\|\, q_{P,i}^*(\theta)) \cdot p(c)\Big[\sum_{\theta} p(x_i|\theta)p^c(\theta)\Big] \\
&= \sum_{x_i}\sum_{c} D_{\mathrm{KL}}(p^c(\theta|x_i) \,\|\, q_{P,i}^*(\theta)) \cdot p(c)\Big[\sum_{\theta} p(x_i|\theta)p^c(\theta)\Big] \\
&= \sum_{x_i} \Big\{ D_{\mathrm{KL}}(p^A(\theta|x_i) \,\|\, q_{P,i}^*(\theta)) \cdot p(c = A)\Big[\sum_{\theta} p(x_i|\theta)p^A(\theta)\Big] + \\
&\qquad\quad D_{\mathrm{KL}}(p^B(\theta|x_i) \,\|\, q_{P,i}^*(\theta)) \cdot p(c = B)\Big[\sum_{\theta} p(x_i|\theta)p^B(\theta)\Big]\Big\}
\end{aligned} \tag{10}
$$

Eq. 10 provides an analytic expression for the information gap for a likelihood-coding population under a task design specified by $(p(c), p^c(\theta))$ and a generative model $p(x|\theta)$. Per observation $x_i$, the expression evaluates the KL divergence between the true posterior $p^c(\theta|x_i)$ and a surrogate posterior $q_{P,i}^*(\theta)$, which is the output of the best possible posterior decoder utilizing the task-marginalized, Bayes-optimal estimator of the prior distribution (Eq. 8). The KL divergence is then marginalized across $x_i$ to derive the total expected performance difference between likelihood decoders and posterior decoders.

### A.1.2 INFORMATION GAP FOR POSTERIOR CODING HYPOTHESIS $\Delta_{\mathrm{P}}^{\mathrm{INFO}}$

For a posterior coding population, the neural population responses $\boldsymbol{r}_{\mathrm{P}}^c$ encode the posterior distribution over $\theta$ given $x$ under the context $c$, i.e. $p^c(\theta|x)$, and are therefore modulated by and dependent on the context prior $p^c(\theta)$:

$$\boldsymbol{r}_{\mathrm{P}}^c \sim f(p^c(\theta|x)), \text{ where } f \text{ is some neural encoding function.}$$

**Decoding performance of a perfect posterior decoder $g_P$**

Applying a posterior decoder $g_P$ to a posterior-coding population $\boldsymbol{r}_\mathrm{P}$, we want

$$g_P(\boldsymbol{r}_\mathrm{P}) \longrightarrow p^c(\theta|x)$$

As before, let us assume the observation space can be discretized into $x \in \{x_i\}$, and consider the neural population responses associated with each $x_i$:

$$\forall x_i, c : \boldsymbol{r}^c_{\mathrm{P},i} \sim f(p^c(\theta|x_i))$$

We denote the output of a posterior decoder as $q^c_{P,i}(\theta) := g_P(\boldsymbol{r}^c_{\mathrm{P},i})$, which is context-dependent as $\boldsymbol{r}^c_{\mathrm{P},i}$ depends on the context $c$. As the output of the posterior decoder $q^c_{P,i}(\theta)$ can be directly interpreted as the posterior distribution (schematized in Fig. 2), the cross-entropy loss for data samples associated with $x_i, c$, i.e. $H(p^c(\theta|x_i), q^c_{P,i}(\theta))$, is minimized when:

$$q^{c*}_{P,i}(\theta) = p^c(\theta|x_i)$$

That is, after training, the posterior decoder output $g_P(\boldsymbol{r}^c_{\mathrm{P},i})$ will converge to $q^{c*}_{P,i}(\theta) = p^c(\theta|x_i)$, provided sufficient training samples are available.

To obtain the expected cross-entropy loss across the entire data set, we marginalize over all $x_i, c$, yielding:

$$\begin{aligned}
\mathbb{E}_{p(x_i,c)}[H(p^c(\theta|x_i), q^{c*}_{P,i}(\theta))] &= \mathbb{E}_{p(x_i,c)}[H(p^c(\theta|x_i)) + D_{\mathrm{KL}}(p^c(\theta|x_i) \,||\, q^{c*}_{P,i}(\theta))] \\
&= \mathbb{E}_{p(x_i,c)}[H(p^c(\theta|x_i))] \\
&= \sum_{x_i,c} H(p^c(\theta|x_i)) \cdot p(x_i, c) \\
&= \sum_{x_i,c} H(p^c(\theta|x_i)) \cdot p(c)\Big[\sum_\theta p(x_i|\theta)p^c(\theta)\Big] \\
&= \sum_{x_i}\sum_c H(p^c(\theta|x_i)) \cdot p(c)\Big[\sum_\theta p(x_i|\theta)p^c(\theta)\Big] \\
&= \sum_{x_i}\Big\{H(p^A(\theta|x_i)) \cdot p(c=A)\Big[\sum_\theta p(x_i|\theta)p^A(\theta)\Big] + \\
&\qquad H(p^B(\theta|x_i)) \cdot p(c=B)\Big[\sum_\theta p(x_i|\theta)p^B(\theta)\Big]\Big\} \qquad (11)
\end{aligned}$$

where the second equality holds because $D_{\mathrm{KL}}(p^c(\theta|x_i) \,||\, q^{c*}_{P,i}(\theta)) = 0$ for a perfect posterior decoder as derived above. Hence, the expected cross-entropy loss for a perfect posterior decoder on a posterior coding population should approach the expected posterior entropy as determined by the context frequencies and context prior distribution as given by Eq. 11. Note Eq. 11 is the same as the expected cross-entropy loss for a perfect likelihood decoder on a likelihood coding population as derived previously in Eq. 7.

**Decoding performance of the best possible likelihood decoder $g_L$**

Applying a likelihood decoder $g_L$ to a posterior coding population $\boldsymbol{r}_\mathrm{P}$, we want

$$g_L(\boldsymbol{r}_\mathrm{P}) \longrightarrow p(x|\theta)$$

In contrast to the mismatched decoding scenario of applying a posterior decoder to a likelihood-coding population where the posterior decoder cannot perfectly decode the posterior distributions from population responses for *any* observation $x_i$, application of a likelihood decoder to a posterior-coding population requires more intricate considerations—we reason below that only *some* $x_i$ would cause the likelihood decoder to fail to perfectly decode the likelihood function from a posterior-coding population. We first reiterate that the posterior population responses $\boldsymbol{r}^c_\mathrm{P}$ are context dependent, which means that for the same $x_i$, the neural responses $\boldsymbol{r}^c_{\mathrm{P},i}$ are different across the two contexts. Hence, from the perspective of a likelihood decoder, for each $x_i$, the inputs (neural responses $\boldsymbol{r}^c_{\mathrm{P},i}$) are different across contexts, but the target output ($p(x_i|\theta)$) is the same. Because the ground-truth context priors $p^c(\theta)$ are explicitly provided to the likelihood decoder, this scenario "pressures"

the decoder to learn a *many-to-one* mapping, which is generally achievable for a sufficiently powerful likelihood decoder (Fig. 2C).

To identify the condition in which the likelihood decoder would fail to perfectly decode the likelihood function from posterior coding population responses, recall that when applying a posterior decoder to a likelihood-coding population, the main reason why the posterior decoder cannot be perfect is that it is forced to map identical inputs ($\boldsymbol{r}_{\mathrm{L},i}$) into multiple distinct target outputs ($p^c(\theta|x_i)$). In other words, the decoder cannot be perfect because it is trying to learn a *one-to-many* mapping. Given this insight, for the scenario of applying a likelihood decoder on a posterior-coding population, we identify the condition under which likelihood decoders are forced to map identical inputs to distinct target outputs. Consider the set of pairs $\chi := \{(x_j, x_k)\}$, where each pair $(x_j, x_k)$ satisfies:

$$\boldsymbol{r}_{\mathrm{P},j}^A \approx \boldsymbol{r}_{\mathrm{P},k}^B$$
$$\Leftrightarrow \quad p^A(\theta|x_j) \approx p^B(\theta|x_k), \ \forall \theta \quad \text{(can be measured in terms of KL divergence)} \qquad (12)$$
$$\Leftrightarrow \quad p^A(\theta) \cdot p(x_j|\theta) \propto p^B(\theta) \cdot p(x_k|\theta), \ \forall \theta$$

That is, we consider the condition $\boldsymbol{r}_{\mathrm{P},j}^A \approx \boldsymbol{r}_{\mathrm{P},k}^B$, where the inputs ($\boldsymbol{r}_{\mathrm{P},j}^A$ or $\boldsymbol{r}_{\mathrm{P},k}^B$) to the likelihood decoder $g_L$ are (approximately) the same but the target output differs based on the context ($p(x_j|\theta)$ or $p(x_k|\theta)$). Under the assumption of ideal decoders, the set of pairs in $\chi = \{(x_j, x_k)\}$ are the only scenarios where it is impossible for an ideal likelihood decoder to be perfect. In these scenarios, identical inputs (population responses encoding the same posterior distributions) need to be decoded into different outputs (distinct likelihood functions), which is not achievable by any functional decoder, regardless of training sample size or expressive of parametrization.

With the insight that only observations in the set of pairs $\chi = (x_j, x_k)$ where Eq. 12 is satisfied will cause the likelihood decoder to fail to perfectly decode the likelihood function, let us now derive the expected likelihood decoder output for each pair. Firstly, consider the frequency of a context given an observation of neural responses associated with $\boldsymbol{r}_{\mathrm{P},j}^A$ or $\boldsymbol{r}_{\mathrm{P},k}^B$:

$$p(c = A|\boldsymbol{r} = \boldsymbol{r}_{\mathrm{P},j}^A \vee \boldsymbol{r}_{\mathrm{P},k}^B) = \frac{p(c = A, \boldsymbol{r} = \boldsymbol{r}_{\mathrm{P},j}^A \vee \boldsymbol{r}_{\mathrm{P},k}^B)}{p(\boldsymbol{r} = \boldsymbol{r}_{\mathrm{P},j}^A \vee \boldsymbol{r}_{\mathrm{P},k}^B)}$$
$$= \frac{p(c = A) \cdot \sum_\theta p^A(\theta)p(x_j|\theta)}{p(c = A) \cdot \sum_\theta p^A(\theta)p(x_j|\theta) + p(c = B) \cdot \sum_\theta p^B(\theta)p(x_k|\theta)}$$

Similarly, we have:

$$p(c = B|\boldsymbol{r} = \boldsymbol{r}_{\mathrm{P},j}^A \vee \boldsymbol{r}_{\mathrm{P},k}^B) = \frac{p(c = B) \cdot \sum_\theta p^B(\theta)p(x_k|\theta)}{p(c = A) \cdot \sum_\theta p^A(\theta)p(x_j|\theta) + p(c = B) \cdot \sum_\theta p^B(\theta)p(x_k|\theta)}$$

Let us denote

$$S_j^A := p(c = A) \sum_\theta p^A(\theta)p(x_j|\theta)$$
$$S_k^B := p(c = B) \sum_\theta p^B(\theta)p(x_k|\theta)$$

Define the observation-dependent context frequency for observing data samples coming from $\boldsymbol{r}_{\mathrm{P},j}^A$ or $\boldsymbol{r}_{\mathrm{P},k}^B$:

$$\rho_j^A := p(c = A|\boldsymbol{r} = \boldsymbol{r}_{\mathrm{P},j}^A \vee \boldsymbol{r}_{\mathrm{P},k}^B) = S_j^A/(S_j^A + S_k^B)$$
$$\rho_k^B := p(c = B|\boldsymbol{r} = \boldsymbol{r}_{\mathrm{P},j}^A \vee \boldsymbol{r}_{\mathrm{P},k}^B) = S_k^B/(S_j^A + S_k^B)$$

Now, let us denote the context-independent likelihood decoder output as $\ell_{jk}(\theta) := g_L(\boldsymbol{r} = \boldsymbol{r}_{\mathrm{P},j}^A \vee \boldsymbol{r}_{\mathrm{P},k}^B)$. The context-dependent posterior distribution given the corresponding context prior $p^c(\theta)$ is

given by:

$$q_{L,j}^A(\theta) = \frac{p^A(\theta)\ell_{jk}(\theta)}{\sum_{\theta'} p^A(\theta')\ell_{jk}(\theta')} = \frac{p^A(\theta)\ell_{jk}(\theta)}{Z_j^A[\ell_{jk}(\theta)]}$$

$$q_{L,k}^B(\theta) = \frac{p^B(\theta)\ell_{jk}(\theta)}{\sum_{\theta'} p^B(\theta')\ell_{jk}(\theta')} = \frac{p^B(\theta)\ell_{jk}(\theta)}{Z_k^B[\ell_{jk}(\theta)]}$$

where $Z_j^A[\ell_{jk}(\theta)]$ and $Z_k^B[\ell_{jk}(\theta)]$ are normalization constants dependent on $\ell_{jk}(\theta)$, defined as:

$$Z_j^A[\ell_{jk}(\theta)] := \sum_\theta p^A(\theta)\ell_{jk}(\theta)$$

$$Z_k^B[\ell_{jk}(\theta)] := \sum_\theta p^B(\theta)\ell_{jk}(\theta)$$

Under cross-entropy loss, we want $\ell_{jk}(\theta)$ (and hence its associated posteriors $q_{L,j}^A(\theta)$ and $q_{L,k}^B(\theta)$) to minimize:

$$\min_{\ell_{jk}(\theta)} \left\{ \rho_j^A H(p^A(\theta|x_j), q_{L,j}^A(\theta)) + \rho_k^B H(p^B(\theta|x_k), q_{L,k}^B(\theta)) \right\}$$

$$= \min_{\ell_{jk}(\theta)} \left\{ -\sum_\theta \left[ \rho_j^A p^A(\theta|x_j) \log q_{L,j}^A(\theta) + \rho_k^B p^B(\theta|x_k) \log q_{L,k}^B(\theta) \right] \right\}$$

$$= \min_{\ell_{jk}(\theta)} \left\{ -\sum_\theta \left[ \rho_j^A \frac{p^A(\theta)p(x_j|\theta)}{\sum_{\theta'} p^A(\theta')p(x_j|\theta')} \log \frac{p^A(\theta)\ell_{jk}(\theta)}{Z_j^A[\ell_{jk}]} + \right.\right.$$

$$\left.\left. \rho_k^B \frac{p^B(\theta)p(x_k|\theta)}{\sum_{\theta'} p^B(\theta')p(x_k|\theta')} \log \frac{p^B(\theta)\ell_{jk}(\theta)}{Z_k^B[\ell_{jk}]} \right] \right\} \tag{13}$$

Define

$$\mu_j^A(\theta) := \rho_j^A p^A(\theta|x_j) = \rho_j^A \frac{p^A(\theta)p(x_j|\theta)}{\sum_{\theta'} p^A(\theta')p(x_j|\theta')} = \frac{p(c=A)p^A(\theta)p(x_j|\theta)}{S_j^A + S_k^B}$$

$$\mu_k^B(\theta) := \rho_k^B p^B(\theta|x_k) = \rho_k^B \frac{p^B(\theta)p(x_k|\theta)}{\sum_{\theta'} p^B(\theta')p(x_k|\theta')} = \frac{p(c=B)p^B(\theta)p(x_k|\theta)}{S_j^A + S_k^B}$$

Note

$$\sum_\theta \mu_j^A(\theta) = \frac{p(c=A)\sum_\theta p^A(\theta)p(x_j|\theta)}{S_j^A + S_k^B} = \rho_j^A$$

$$\sum_\theta \mu_k^B(\theta) = \frac{p(c=B)\sum_\theta p^B(\theta)p(x_k|\theta)}{S_j^A + S_k^B} = \rho_k^B$$

The cross-entropy loss term in Eq. 13 can be rewritten as:

$$L(\ell_{jk}(\theta)) = -\sum_\theta \left[ \mu_j^A(\theta) \cdot \left( \log p^A(\theta) + \log \ell_{jk}(\theta) - \log Z_j^A[\ell_{jk}(\theta)] \right) + \right.$$

$$\left. \mu_k^B(\theta) \cdot \left( \log p^B(\theta) + \log \ell_{jk}(\theta) - \log Z_k^B[\ell_{jk}(\theta)] \right) \right]$$

$$= -\left\{ \sum_\theta \left[ \mu_j^A(\theta) \log p^A(\theta) + \mu_k^B(\theta) \log p^B(\theta) \right] \right.$$

$$+ \sum_\theta \left[ (\mu_j^A(\theta) + \mu_k^B(\theta)) \cdot \log \ell_{jk}(\theta) \right]$$

$$\left. - \left[ \sum_\theta \mu_j^A(\theta) \right] \cdot \log Z_j^A[\ell_{jk}(\theta)] - \left[ \sum_\theta \mu_k^B(\theta) \right] \cdot \log Z_k^B[\ell_{jk}(\theta)] \right\} \tag{14}$$

Note from Eq. 14, we can see that $L(\alpha\ell) = L(\ell)$, $\forall \alpha > 0$, as the normalization factors cancel out the multiplicative effect. Therefore $\ell^*$ that minimizes $L$ is determined up to a multiplicative constant, agreeing with our intuition that the output of a likelihood decoder should be only determined up to a multiplicative constant as in Eq. 6.

The above minimization happens at the critical point $\ell_{jk}^*(\theta)$ where $\frac{\partial L}{\partial \ell_{jk}^*(\theta)} = 0$, $\forall \theta$, with $L$ defined in Eq. 14.

Before proceeding to find the minimum for this variational calculus problem, let us first evaluate:

$$\frac{\partial}{\partial \ell_{jk}(\theta)} Z_j^A[\ell_{jk}(\theta)] = \frac{\partial}{\partial \ell_{jk}(\theta)} \left\{ \sum_{\theta'} p^A(\theta') \ell_{jk}(\theta') \right\} = p^A(\theta)$$

$$\frac{\partial}{\partial \ell_{jk}(\theta)} Z_k^B[\ell_{jk}(\theta)] = \frac{\partial}{\partial \ell_{jk}(\theta)} \left\{ \sum_{\theta'} p^B(\theta') \ell_{jk}(\theta') \right\} = p^B(\theta)$$

To find the minimum , let us take the derivative of $L$ with respect to $\ell_{jk}(\theta)$ and set it to zero:

$$0 = \frac{\partial L(\ell_{jk}(\theta))}{\partial \ell_{jk}(\theta)}$$

$$= -\frac{\partial}{\partial \ell_{jk}(\theta)} \Big\{ \sum_{\theta} \left[ \mu_j^A(\theta) \log p^A(\theta) + \mu_k^B(\theta) \log p^B(\theta) \right]$$

$$+ \sum_{\theta} \left[ \left( \mu_j^A(\theta) + \mu_k^B(\theta) \right) \cdot \log \ell_{jk}(\theta) \right]$$

$$- \left[ \sum_{\theta} \mu_j^A(\theta) \right] \cdot \log Z_j^A[\ell_{jk}(\theta)] - \left[ \sum_{\theta} \mu_k^B(\theta) \right] \cdot \log Z_k^B[\ell_{jk}(\theta)] \Big\}$$

$$= -\left\{ \frac{\mu_j^A(\theta) + \mu_k^B(\theta)}{\ell_{jk}(\theta)} - \frac{\left[ \sum_{\theta} \mu_j^A(\theta) \right]}{Z_j^A[\ell_{jk}(\theta)]} \frac{\partial Z_j^A[\ell_{jk}(\theta)]}{\partial \ell_{jk}(\theta)} - \frac{\left[ \sum_{\theta} \mu_k^B(\theta) \right]}{Z_k^B[\ell_{jk}(\theta)]} \frac{\partial Z_k^B[\ell_{jk}(\theta)]}{\partial \ell_{jk}(\theta)} \right\}$$

$$= -\left\{ \frac{\mu_j^A(\theta) + \mu_k^B(\theta)}{\ell_{jk}(\theta)} - \frac{\rho_j^A}{Z_j^A[\ell_{jk}(\theta)]} p^A(\theta) - \frac{\rho_k^B}{Z_k^B[\ell_{jk}(\theta)]} p^B(\theta) \right\}$$

Therefore the minimization happens when (determined up to a multiplicative constant):

$$\ell_{jk}^*(\theta) \propto \frac{\mu_j^A(\theta) + \mu_k^B(\theta)}{\frac{\rho_j^A}{Z_j^A[\ell_{jk}^*]} p^A(\theta) + \frac{\rho_k^B}{Z_k^B[\ell_{jk}^*]} p^B(\theta)}$$

$$= \frac{\rho_j^A p^A(\theta|x_j) + \rho_k^B p^B(\theta|x_k)}{\frac{\rho_j^A}{Z_j^A[\ell_{jk}^*]} p^A(\theta) + \frac{\rho_k^B}{Z_k^B[\ell_{jk}^*]} p^B(\theta)} \tag{15}$$

Eq. 15 gives an implicit expression for $\ell_{jk}^*(\theta)$, since both $Z_j^A[\ell_{jk}^*]$ and $Z_k^B[\ell_{jk}^*]$ depend on $\ell_{jk}^*(\theta)$. The equation can be solved using fixed-point iteration starting with some initial guess for $\ell_{jk}^{(0)}(\theta) > 0$. For instance:

Initialize $\ell_{jk}^{(0)}(\theta) \propto 1$

for $t = 0, 1, 2, \dots$ :

compute $Z_j^{A,(t)}[\ell_{jk}^{(t)}] = \sum_{\theta} \ell_{jk}^{(t)}(\theta) p^A(\theta)$

$$Z_k^{B,(t)}[\ell_{jk}^{(t)}] = \sum_{\theta} \ell_{jk}^{(t)}(\theta) p^B(\theta)$$

update $\ell_{jk}^{(t+1)}(\theta) = \frac{\rho_j^A p^A(\theta|x_j) + \rho_k^B p^B(\theta|x_k)}{\frac{\rho_j^A}{Z_j^{A,(t)}[\ell_{jk}^{(t)}]} p^A(\theta) + \frac{\rho_k^B}{Z_k^{B,(t)}[\ell_{jk}^{(t)}]} p^B(\theta)}$

Stop when $\ell_{jk}^{(t)}(\theta)$ converges (up to a multiplicative constant).

That is, as given by Eq. 15, after training, the best possible likelihood decoder output for data samples associated with $r_{\mathrm{P},j}^A$ and $r_{\mathrm{P},k}^B$ is as if the likelihood decoder were to divide a surrogate posterior that is a weighted sum of ground-truth posteriors $\rho_j^A p^A(\theta|x_j) + \rho_k^B p^B(\theta|x_k)$ by a surrogate prior that is a weighted sum of ground-truth priors $\frac{\rho_j^A}{Z_j^A[\ell_{jk}^*]} p^A(\theta) + \frac{\rho_k^B}{Z_k^B[\ell_{jk}^*]} p^B(\theta)$.

The posterior of the best possible likelihood decoder output $g_L^* = \ell_{jk}^*(\theta)$ given the corresponding context prior for $r_{\mathrm{P},j}^A$ and $r_{\mathrm{P},k}^B$ is evaluated as:

$$q_{L,j}^{A*}(\theta) = \frac{\ell_{jk}^*(\theta) p^A(\theta)}{Z_j^A[\ell_{jk}^*]}$$

$$q_{L,k}^{B*}(\theta) = \frac{\ell_{jk}^*(\theta) p^B(\theta)}{Z_k^B[\ell_{jk}^*]}$$

Hence, to obtain the expected cross-entropy loss across the entire data set, we marginalize over all $x_i, c$, and the total cross-entropy loss for the best possible likelihood decoder can be expressed as:

$$\begin{aligned}
\mathbb{E}_{p(x_i,c)}[H(p^c(\theta|x_i), q_{L,i}^{c*}(\theta))] &= \mathbb{E}_{p(x_i,c)}[H(p^c(\theta|x_i)) + D_{\mathrm{KL}}(p^c(\theta|x_i) \| q_{L,i}^{c*}(\theta))] \\
&= \mathbb{E}_{p(x_i,c)}[H(p^c(\theta|x_i))] + \mathbb{E}_{p(x_i,c)}[D_{\mathrm{KL}}(p^c(\theta|x_i) \| q_{L,i}^{c*}(\theta))] \\
&= \text{CE loss for the perfect posterior decoder (Eq. 11)} \\
&\quad + \mathbb{E}_{p(x_i,c)}[D_{\mathrm{KL}}(p^c(\theta|x_i) \| q_{L,i}^{c*}(\theta))]
\end{aligned} \tag{16}$$

**Information gap for a posterior coding population $\Delta_{\mathrm{P}}^{\mathrm{info}}$**

From equation 16, let us define $\Delta_{\mathrm{P}}^{\mathrm{info}}$, the information gap for a posterior coding population between a perfect posterior decoder ($g_P^*$) and the best possible likelihood decoder ($g_L^*$), as the expected difference in the cross-entropy loss of the two decoders:

$$\begin{aligned}
\Delta_{\mathrm{P}}^{\mathrm{info}} &:= \mathbb{E}_{p(x_i,c)}[D_{\mathrm{KL}}(p^c(\theta|x_i) \| q_{L,i}^{c*}(\theta))] \\
&= \sum_{x_i,c} D_{\mathrm{KL}}(p^c(\theta|x_i) \| q_{L,i}^{c*}(\theta)) \cdot p(x_i,c), \quad \text{since only } x_i \in \chi = \{(x_j, x_k)\} \text{ terms are nonzero} \\
&= \sum_{x_i \in \chi, c} D_{\mathrm{KL}}(p^c(\theta|x_i) \| q_{L,i}^{c*}(\theta)) \cdot p(x_i,c) \\
&= \sum_{x_i \in \chi, c} D_{\mathrm{KL}}(p^c(\theta|x_i) \| q_{L,i}^*(\theta)) \cdot p(c) \Big[ \sum_\theta p(x_i|\theta) p^c(\theta) \Big] \\
&= \sum_{x_i \in \chi} \sum_c D_{\mathrm{KL}}(p^c(\theta|x_i) \| q_{L,i}^{c*}(\theta)) \cdot p(c) \Big[ \sum_\theta p(x_i|\theta) p^c(\theta) \Big] \\
&= \sum_{(x_j, x_k)} \Big\{ D_{\mathrm{KL}}(p^A(\theta|x_j) \| q_{L,j}^{A*}(\theta)) \cdot p(c=A) \Big[ \sum_\theta p(x_j|\theta) p^A(\theta) \Big] \\
&\quad + D_{\mathrm{KL}}(p^B(\theta|x_k) \| q_{L,k}^{B*}(\theta)) \cdot p(c=B) \Big[ \sum_\theta p(x_k|\theta) p^B(\theta) \Big] \Big\}
\end{aligned} \tag{17}$$

Eq. 17 provides an analytic expression for the information gap for a posterior-coding population under a task design specified by $(p(c), p^c(\theta))$ and a generative model $p(x|\theta)$. Per pair of observations $(x_j, x_k)$, we evaluate the KL divergence between the true posterior ($p^A(\theta|x_j)$ or $p^B(\theta|x_k)$) and a surrogate posterior ($q_{L,j}^{A*}(\theta)$ or $q_{L,k}^{B*}(\theta)$), which is the posterior distribution associated with the output of the best possible likelihood decoder utilizing the task-marginalized, Bayes-optimal estimators as given by Eq. 15. The KL divergence is then marginalized across the pairs $(x_j, x_k)$ to derive the total expected performance difference between likelihood decoders and posterior decoders.

## A.2 SCHEMATICS FOR TASK DESIGN TRADEOFF

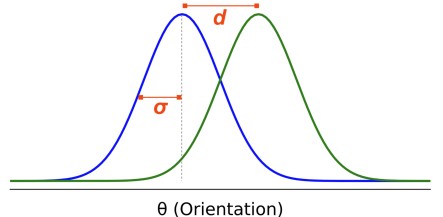

Figure 8: **Design tradeoff.**

## A.3 DETAILS OF SIMULATION EXPERIMENT

### A.3.1 SIMULATED NEURAL POPULATIONS

We consider tasks with Gaussian context priors motivated by classic orientation discrimination psychophysical tasks (Orbán et al., 2016; Walker et al., 2020). In this task, subjects perform an orientation discrimination under two contexts $c \in \{A, B\}$, with the context for each session sampled randomly, i.e. $p(c = A) = p(c = B) = 0.5$. Within each session, the trial-to-trial hidden world state $\theta$ (i.e. orientation) is drawn from context-specific Gaussian prior distributions $p^c(\theta) = \mathcal{N}(\mu^c, (\sigma^c)^2)$, where $\mu^c$ and $(\sigma^c)^2$ are task-specific parameters. In the simulation, we consider $\theta \in \{-90°, 90°\}$, and use identical variances for the two Gaussian priors $\sigma^A = \sigma^B = \sigma$. Consequently, the experimental design is fully specified by the tuple of task parameters $(\mu^A, \mu^B, \sigma)$. Furthermore, foregoing cardinal orientation consideration, the circular symmetry of orientations $\theta$ suggests that only the separation between the two means $d = |\mu^A - \mu^B|$ would meaningfully impact perception. Given this, we always center the two means around zero, meaning $\mu^A = -\frac{1}{2}d$ and $\mu^B = \frac{1}{2}d$. We systematically vary $(d, \sigma)$ to cover the task spectrum of Gaussian context priors in the simulation studies.

Noisy sensory observations $x$ are drawn from the conditional distribution defined by the given generative model $p(x|\theta)$. This stochastic process can be seen as capturing both intrinsic neuronal noise and uncertainty in the extrinsic stimulus features. This generative model can be experimentally manipulated through stimulus parameters such as contrast, where lower contrast induces increased observation variance, reflecting increased sensory uncertainty. In the simulation, $p(x|\theta)$ is modeled as Gaussian distributions to reflect Gaussian orientation tuning curves commonly found among simple V1 neurons. We model the effect of different contrast levels by systematically varying the standard deviation of the generative model $\sigma_{\text{obs}}$ (Walker et al., 2020). To this end, standard deviations $\sigma_{\text{obs}}$ of 8, 15, and 25 are chosen to model the generative model under high, medium, and low contrast levels, respectively. Finally, on each trial, the hidden world state $\theta$ is drawn from $p^c(\theta) = \mathcal{N}(\mu^c, (\sigma^c)^2)$ and then the observation is drawn from the conditional distribution $p(x|\theta) = \mathcal{N}(\theta, \sigma_{\text{obs}}^2)$.

For simulated population responses, we first implement Poisson neuron models with Gaussian tuning curves and Poisson variability (Walker et al., 2020). A population of neurons indexed by $l$, ranging from 5-500 neurons, was constructed with Gaussian tuning curves $\mathcal{N}(\theta_l, \sigma_{\text{obs}}^2)$ with their means $\theta_l$ tiling up the orientation space and their standard deviations being $\sigma_{\text{obs}}$. For likelihood-coding populations, the mean firing rate of each neuron on each trial, after an observation $x$ is sampled, is determined by the probability density of its Gaussian tuning curve, i.e. $f(x) = \frac{1}{\sqrt{2\pi\sigma_{\text{obs}}^2}}e^{-\frac{(x-\theta_l)^2}{2\sigma_{\text{obs}}^2}}$, scaled with a fixed constant of 30 to approximate the typical range of neuron firing rates observed experimentally (Walker et al., 2020). For posterior-coding populations, the mean firing rate of each neuron is further multiplied by the context-specific prior $p^c(\theta)$, thus effectively encoding the posterior $p^c(\theta|x) \propto p(x|\theta) \cdot p^c(\theta)$ in their mean firing rates. For both populations, trial-to-trial spike counts are then generated by sampling from Poisson distribution with the specified mean firing rates.

For some simulation experiments, we additionally implemented a more complex, gain-modulated Poisson neuron model for simulating population responses (Goris et al., 2014). The gain-modulated Poisson neuron model has been proposed to account for the supra-Poisson variability commonly observed experimentally among V1 neurons. In this model, the mean firing rate of the neuron is

the product of two terms: 1) the original rate determined by the Gaussian tuning curve model, and 2) a stimulus-independent gain factor $G$. Goris et al. (2014) proposed and validated on V1 neural data that this stimulus-independent gain factor $G$ can be effectively modeled as following a gamma distribution with a mean of one and variance of $\sigma_G^2$. Based on their results, we choose a biologically realistic value of $\sigma_G \approx 0.5$ in our simulation. Therefore, on a trial-to-trial basis, after an original rate is determined according to the procedure in the previous paragraph for the likelihood-coding or posterior-coding population, a random gain factor is then sampled from the gamma distribution and multiplied with the original rate to get the mean firing rate for the gain-modulated Poisson neuron model. Similarly, spike counts are then generated from the mean firing rates with Poisson variability.

### A.3.2    EXAMPLE LIKELIHOOD AND POSTERIOR

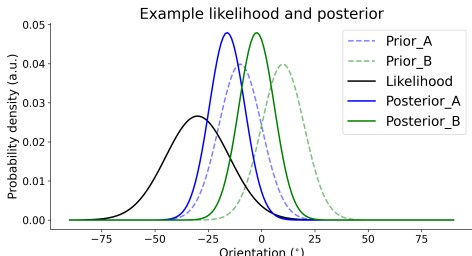

Figure 9: Example of the ground truth likelihood, priors, and posteriors.

### A.3.3    PROBABILISTIC INFORMATION DECODER

As described in Fig. 2C, deep neural networks parametrized by multi-layered perceptrons are trained with cross-entropy loss to serve as flexible, powerful decoders to decode either the likelihood function or the posterior distribution from simulated neural population responses (Walker et al., 2020). We use fully-connected, deep neural networks with two hidden layers, with 300 and 200 units in the first and second layer, respectively. All hidden units are rectified linear units, and dropout rates of 0.5 are used for both layers. The input dimension to the first layer is the number of neurons in the simulated population, ranging from 5–500. The output layer is a fully connected readout with no nonlinearity and a dimension of the number of possible hidden states. In our simulation, we consider orientation $\theta \in \{-90°, 90°\}$ and discretize them into one degree bins, leading to a total number of possible hidden states of 181. To facilitate numerical stability, the decoded probability quantity is operating in the log space. The posterior decoder output is treated as the log-posterior, which is directly optimized to minimize the cross-entropy loss. The likelihood decoder output is treated as the log-likelihood, which is then integrated with the ground truth log-prior to arrive at the final output that is optimized to minimize the cross-entropy loss. To encourage smoothness of the decoded probability distributions, an $L_2$ regularizer on the log-posteriors filtered with a Laplacian filter of the form $h = [0.25, 0.5, 0.25]$ is added to the cross-entropy term, as proposed in (Walker et al., 2020). We use $(0.8, 0.2)$ for train-validation split for training the decoders A held-out test set is used to final evaluation and all results in the paper are on the test test. Early stop with patience of 10 and minimal change of 2e-6 in validation set cross-entropy loss is adopted to prevent overfitting. All models were constructed and trained using the Pytorch framework (Paszke et al., 2019).

### A.4 INCORPORATING BIASED PRIOR FROM BEHAVIOR DATA

Although our main analysis focuses on the theoretical decoding limit using the optimal priors, our framework naturally accommodates model mismatch or biased priors by incorporating behavioral (psychometric) measurements. The procedure is detailed in Fig. 10:

- **Analyze the psychometric curve**: In perceptual tasks (e.g., orientation discrimination tasks), deviations in the subject's psychometric curve (correct rate as a function of stimulus orientation) from the ideal observer reveal model mismatch or biased priors.

- **Infer the subject's model mismatch/ biased prior**: Features (such as leftward shifts or increased slope/variance) in the psychometric function can be mapped to corresponding biases or increased uncertainty in the subject's internal mismatched prior.

- **Compute the information gap using the inferred prior**: The inferred biased prior can then be used directly in our information-gap calculation, yielding predictions that account for the subject's model mismatch and more accurately reflect expected empirical decoder differences.

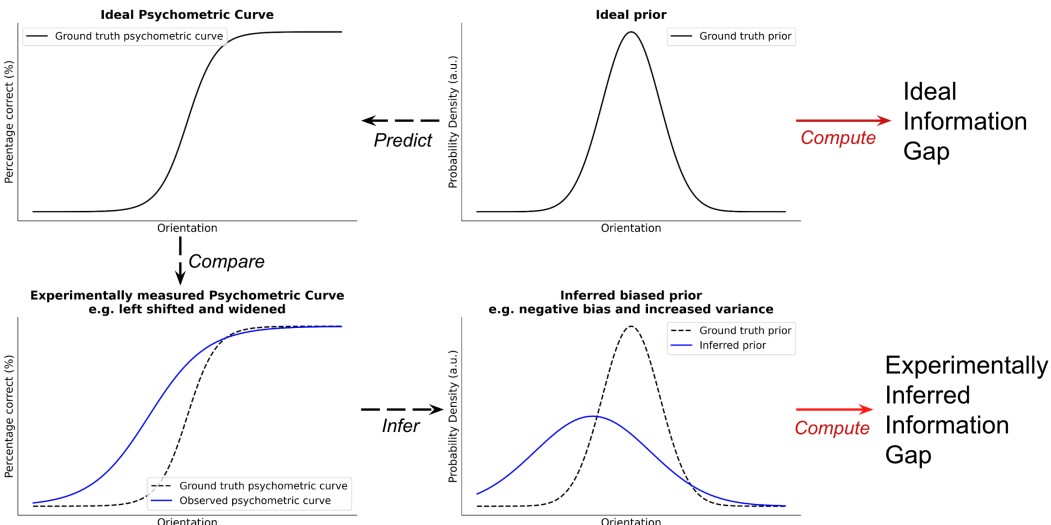

Figure 10: The information gap computation can incorporate behavior data by estimating the subject's biased prior from its psychometric curve.

### A.5 EXTENDING OUR FRAMEWORK TO MIXED CODING HYPOTHESIS

In this section we discuss a more nuanced probabilistic coding hypothesis and how our proposed framework could be extended to identify or falsify it. As an example of *mixed* coding hypothesis, Ganguli & Simoncelli (2010; 2014) proposed combing heterogeneous tuning curves—which embed aspects of the prior—with spiking variability that reflects the likelihood, yielding a hybrid code in which sensory responses carry both likelihood information and a structurally instantiated prior. This example can be categorized as a mixed or intermediate hypothesis, in between the canonical pure likelihood and pure posterior coding hypothesis. Our framework can naturally accommodate such mixed coding hypothesis by evaluating how each decoder performs under mismatched information. As shown in Fig. 11, since now both the likelihood and posterior decoders can recover the correct distributions, our theory predicts an information gap $\Delta^{\text{info}} = 0$. This zero-info-gap signature is distinct and does not arise under optimized task designs for either pure likelihood- or pure posterior-coding populations, which produce reliably nonzero and separable values. As a result, optimizing the task to maximally separate the two canonical hypotheses simultaneously maximizes sensitivity to departures from them. A mixed code that yields $\Delta^{\text{info}} = 0$ under the same optimized design becomes cleanly identifiable as neither pure likelihood nor pure posterior. Thus this discussion illustrates how our method could generalize beyond the two extreme hypotheses and provides a principled tool for distinguishing both pure and mixed coding schemes.

More broadly, we do not claim that likelihood and posterior coding are the only relevant theories in the literature, but they represent the two major families of theories that differ in what probabilistic quantity is encoded. Our contribution is to provide a principled methodology for experimentally distinguishing such theories. By optimizing task parameters to maximally separate these canonical extremes, we simultaneously maximize sensitivity to discriminating more nuanced probabilistic coding theories like the mixed coding hypothesis.

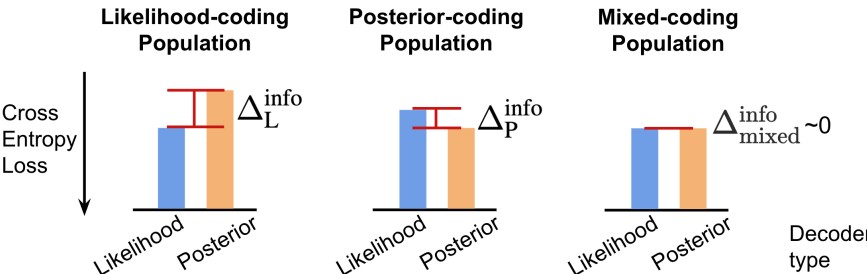

Figure 11: Under mixed coding hypothesis, the information gap becomes zero.

### A.6 EFFECT OF NOISE AND FIRING RATE

As shown in Fig. 12, decreasing firing rates or increasing noise slows the convergence of empirical decoder performance differences. More trials are needed for the empirical decoder performance difference to approach the theoretical information gap. However, with sufficient data, the decoder performance differences ultimately converge to the same theoretical value. This reflects the expected effect of reduced signal-to-noise ratio—decoding becomes harder, but the underlying difference in decodable information is unchanged. Thus, while low SNR increases data requirements, the theoretical information gap remains the correct predictor of the asymptotic difference between the two hypotheses.

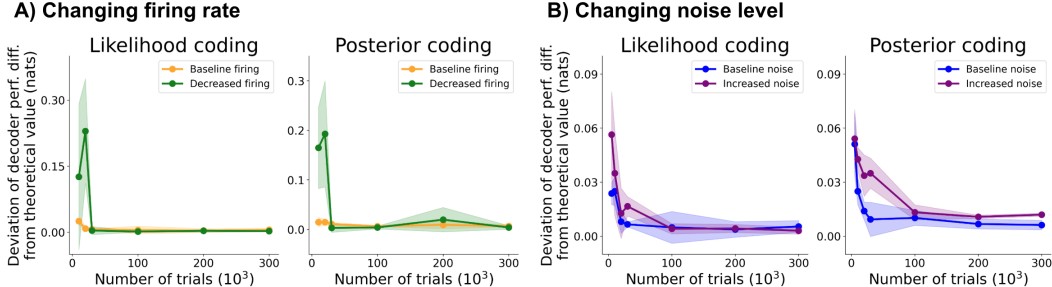

Figure 12: Effect of firing rate and noise level.

### A.7 FACTORS AFFECTING CONVERGENCE SPEED

We conducted ablation studies to examine the factors that determine how quickly empirical performance converges to the theoretical value of information gap. Our main simulations assume that neural tuning curves tile the full orientation space, consistent with standard V1 models Rubin et al., 2015. In Fig. 13A, when the population is randomly sampled without full coverage of the entire orientation space, since no decoder can recover information about orientations lacking tuned neurons, we found that convergence with respect to neuron count becomes substantially slower. In addition, in Fig. 3, the neuron-scaling experiment uses 30k trials so that decoders quickly approach the theoretical limit. In Fig. 13B, we performed an ablation with fewer trials (3k trials) and observed that convergence is again slower because the decoder cannot reliably estimate the encoded distributions from limited data. In practice, the above factors can be mitigated by modern neurophysiology population recordings that provide large number of trials with hundreds to thousands of simultaneously recorded neurons that cover full range of orientation space.

Finally, to demonstrate that our result is robust to the level of discretization of the orientation variable, we repeated the convergence analysis with higher-resolution orientation bins (0.25° instead of 1° as in the main results), and obtained indistinguishable results. This confirms that the accuracy of information gap and its empirical convergence are robust to binning and reflect the underlying decodable information rather than numerical artifacts.

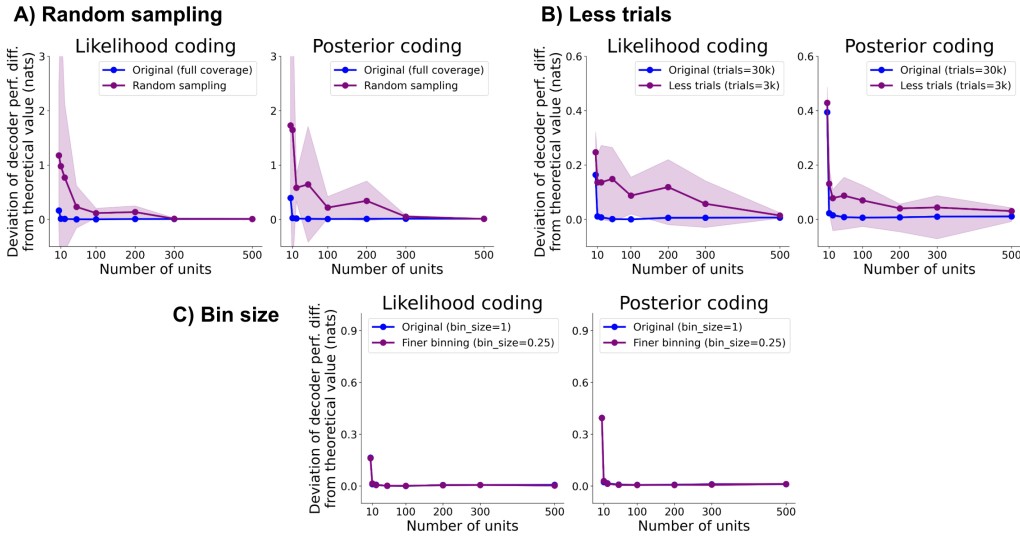

Figure 13: Examine factors affecting convergence speed. A) Effect of orientation coverage. B) Effect of trial numbers. C) Effect of bin size.

## A.8 DETAILED RESULTS ON HEAVY-TAILED PRIORS

Below we first provide the full results of information gap landscape across various contrast levels for heavy-tailed context priors including student's t-distribution (Fig. 14) and Cauchy distribution (Fig. 15). Note that for t-distribution we report the results using degrees of freedom $\nu = 3$. When $\nu \to \infty$ the t-distribution reduces to a standard Gaussian distribution, and when $\nu = 0$ the t-distribution becomes the Cauchy distribution. We then provide an intuitive example explaining why the information gap for posterior coding hypothesis is dramatically lower under heavy-tailed context priors compared to Gaussian context priors. The main reason is that under Gaussian generative models, when integrated with heavy-tailed priors, the posteriors tend to become asymmetric (as opposed to Gaussian priors where the posteriors are still symmetric Gaussian), thus limiting the number of pairs $(x_j, x_k)$ that could confuse the likelihood decoder.

### A.8.1 INFORMATION GAP LANDSCAPE

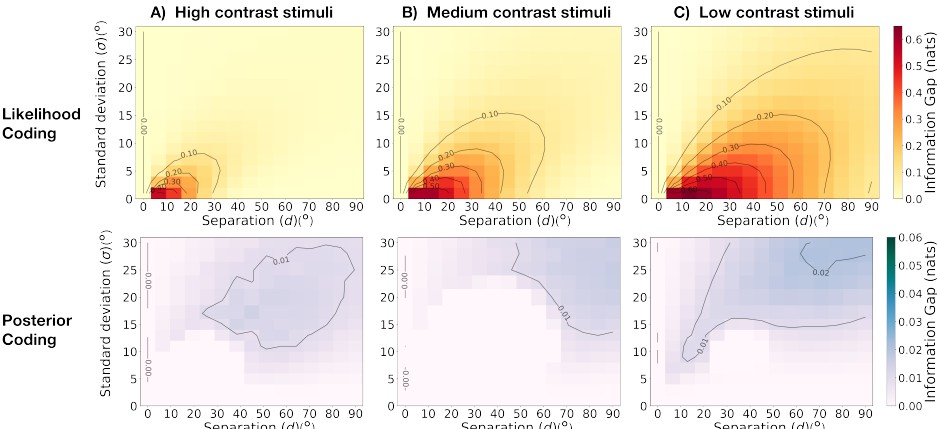

Figure 14: **Information gap landscapes when using student's t-distribution with degrees of freedom $\nu = 3$ as context priors.** A) Information gap as a function of task parameters ($d$: separation between context priors, and $\sigma$: context prior standard deviations) for both the likelihood coding hypothesis (top) and the posterior coding hypothesis (bottom) when presented with high contrast stimuli. B) Same for medium contrast stimuli and C) for low contrast stimuli.

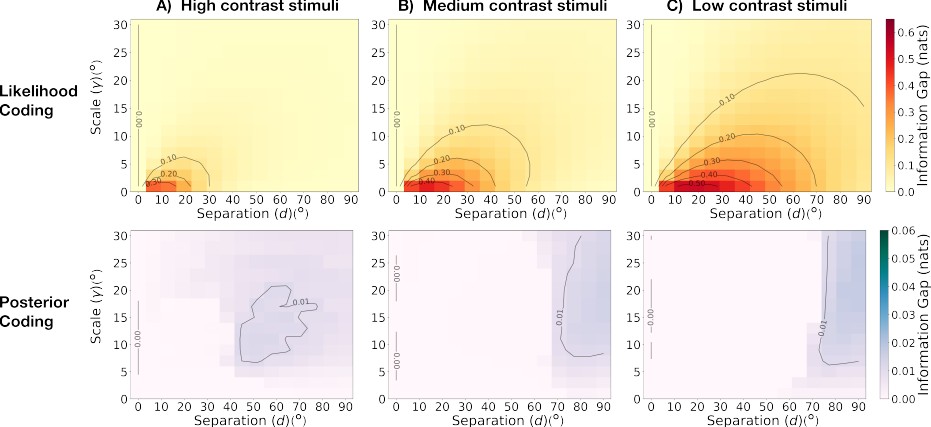

Figure 15: **Information gap landscapes when using Cauchy distribution as context priors.** A) Information gap as a function of task parameters ($d$: separation between context priors, and $\gamma$: context prior scales) for both the likelihood coding hypothesis (top) and the posterior coding hypothesis (bottom) when presented with high contrast stimuli. B) Same for medium and C) low contrast stimuli.

### A.8.2 An example explaining why $\Delta_{\text{P}}^{\text{INFO}}$ is dramatically decreased under heavy-tailed context priors

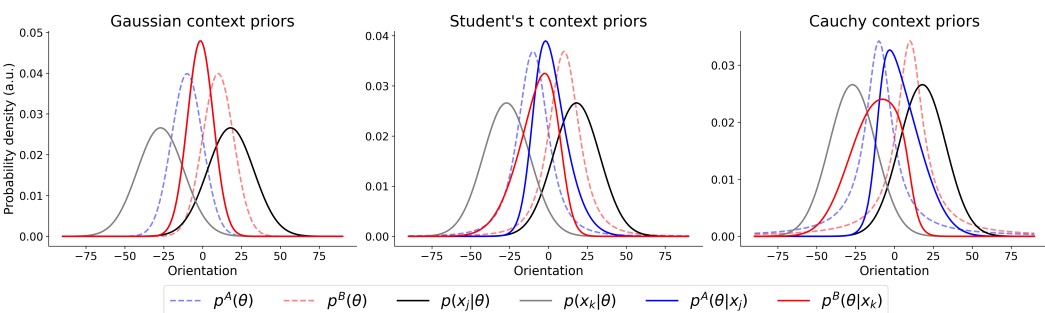

Figure 16: **Heavy tailed context priors, when integrated with Gaussian likelihood function, lead to asymmetric posterior distributions, limiting the pairs of identical posteriors satisfying Eq. 12 that would cause imperfect likelihood decoders on posterior-coding populations.** Across task designs with Gaussian context priors (left), student's t context priors with $\nu = 3$ (middle), and Cauchy context priors (right), the context priors $p^A(\theta)$ and $p^B(\theta)$ are shown in dashed blue and red lines, respectively. Note they all share identical standard deviation or scale parameters to facilitate comparison. One example pair of $(x_j, x_k) = (18°, -27°)$ that satisfies Eq. 12 under Gaussian context priors is shown here, with the associated likelihood functions $p(x_j|\theta)$ and $p(x_k|\theta)$ plotted in solid gray and black lines, respectively. The posterior distributions under each context priors, $p^A(\theta|x_j)$ and $p^B(\theta|x_k)$ are shown as solid blue and red lines, respectively. Under Gaussian context priors (left), the two posteriors are equal to each other, i.e. $p^A(\theta|x_j) = p^B(\theta|x_k)$, hence the two lines overlap. However, as the context priors become increasingly heavy-tailed as under student's t distribution (middle) and Cauchy distributions (right), the two posteriors become more and more asymmetric, leading to non-identical posterior distributions that no longer satisfy Eq. 12. This example demonstrates why there are much less pairs $(x_j, x_k)$ that would satisfy Eq. 12, accounting for the observation that the information gap of posterior-coding population is dramatically decreased under heavy-tailed context priors.

### A.9 RESULTS ON THIN-TAILED PRIORS

To provide further examples on non-Gaussian context priors, we examined thin-tailed distributions as stimulus prior distributions. We reported additional analyses using canonical thin-tailed generalized normal distributions with $\beta > 2$) in Fig. 17. The information gap landscape shows that thin-tailed priors similarly lead to near-0 posterior-coding information gaps across task parameter space. Our framework provides a similar explanation: under thin-tailed context priors, the resulting posteriors become highly asymmetric across contexts (Fig. 18), reducing the feasible set of $(x_j, x_k)$ pairs that can satisfy Eq. 12, thereby shrinking the posterior-coding information gap, which mirrors the failure mode observed with heavy-tailed priors.

What about uniform priors? As shown in Fig. 18, a uniform prior induces no context-dependent modulation of the posterior. Hence, likelihood- and posterior-coding populations become nearly indistinguishable, causing the information gap to collapse for both hypotheses.

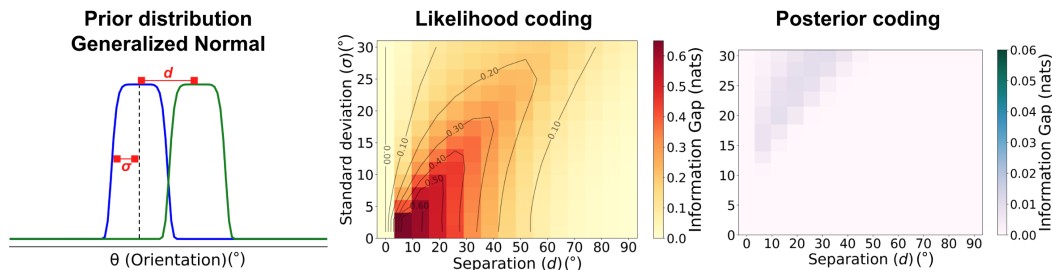

Figure 17: **Information gap landscapes when using generalized normal distribution as context priors.** A) Information gap as a function of task parameters ($d$: separation between context priors, and $\sigma$: context prior standard deviations) for both the likelihood coding hypothesis (middle) and the posterior coding hypothesis (right).

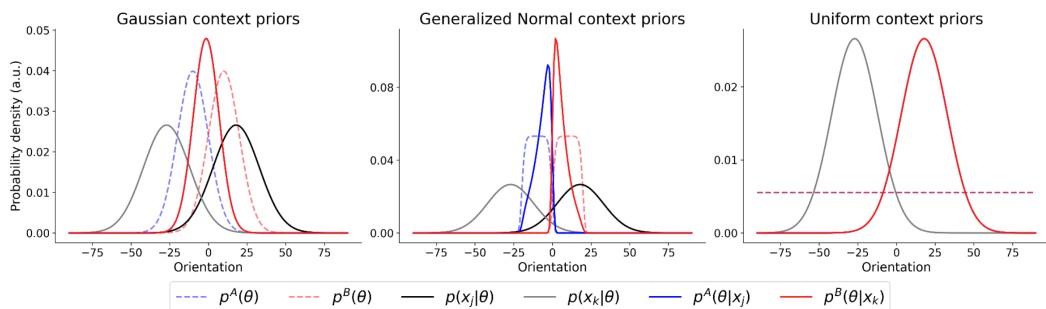

Figure 18: Thin tailed context priors, when integrated with Gaussian likelihood function, lead to asymmetric posterior distributions, limiting the pairs of identical posteriors satisfying Eq. 12 that would cause imperfect likelihood decoders on posterior-coding populations.

