# OpenReview forum: "An Information-Theoretic Framework For Optimizing Experimental Design To Distinguish Probabilistic Neural Codes"
_ICLR.cc/2026/Conference — ICLR 2026 Poster_

### Official Review · Reviewer_JDGu · 2025-10-30

**Soundness:** 3
**Presentation:** 2
**Contribution:** 3
**Rating:** 6
**Confidence:** 4

**Summary:**

The paper puts forth a mathematical formalism for experimentally distinguishing between encoding of likelihood or of posterior at the level of neural populations via decoding, provides an information gap metric for the discriminability of the two, and optimizes in the space of tasks for designs which most clearly separate between the two hypotheses. This paves the way for a more precise experimental validation  of various bayesian accounts of perception and associated neural implementation in the brain.

The basic setup assumes a family of tasks where the animal needs to estimate and report a one-dimensional latent variable theta conveyed by a sensory stimulus, in two (known to the animal) contexts corresponding to different theta distributions. given the responses of a population of sensory neurons r, from which different probabilistic information (according to an experimentally matched ideal observer) is decoded via flexible deep learning models. Assumptions about the different models are incorporated in the network's architecture following Walker et al 2020. The information gap between them is the KL between the two possible encoded quantities (using discretized random variables for easy computation). The question then becomes how to optimize the distance and degree of overlap for priors in the two contexts to maximize discriminability

**Strengths:**

Interesting question, given precise mathematical form.

Novelty and significance: both likelihood and posterior hypotheses make qualitatively similar predictions in traditional experimental designs which has hampered the ability to experimentally adjudicate between probabilistic coding theories. New ways to design experiments for maximal discriminability would circumvent this problem. To my knowledge no formal framing for such experimental design exists to date.

the idea that crossentropy-based decoders trained on ground truth stimulus identity are linked to ideal observer posterior uncertainty for the same data distribution seems generally useful

Numerical validation of theoretical results in the limit of enough data

**Weaknesses:**

While i think the general framework is sensible and fruitful, I have concerns about the details of the execution.

Although the writing overall is overall pretty clear, the key text with the description of the information gap is difficult to parse and ambiguous. the logic of the construction is not clearly enough explained in that part of the text.
The appendix explains it better that the goal is to link crossentropy of different decoders to ideal observer posteriors arguing that a mismatched decoder will achieve overall less crossentropy performance but the main text cannot really be understood on its own

But the weakness is that the punchline is rather weak: the results do not seem to provide much insight into how one could design better experiments beyond giving ranges of possible d' parameters that look good c.f. the theory. It is not clear if the (not negligible) experimental data collected in this setup is outside that regime so that experimentalists would know to do something differently.

**Questions:**

How does the animal's own added sensory uncertainty get incorporated in the procedure ? can the (unavoidable) model mismatch bias the conclusions in any way or are there formal guarantees that it will not?

In particular, what if the local neural population encodes beliefs about related random variables but not the exact parametrization of the experiment? I am mentioning this because successful neural sampling (posterior-based) probabilistic accounts of V1 responses use a high dimensional distributed representation (think probabilistic versions of factor analysis or gaussian scale mixtures) related to orientation but not beliefs about theta per se

i got a little confused about terminology, e.g. "true posterior ...of a ideal likelihood decoder"?  i think what is meant is an optimal decoder of likelihood or posterior beliefs from neural responses

As defined, the information gap seems to come from the ability of the agent to correctly know or infer contex, some more discussion on that would be helpful

Please explain the need for hypothesis specific information gaps, what assumptions it makes about context unknown r, and what it implies for experimental design: do you assume data exists for both a single/cued context version and an inferred/uncued context version of the same task? The elements of the math on the bounds of decoder crossentropy performance described in A1 are individually more comprehensible but the big picture logic and assumptions remain to be inferred between the lines

Eq3: i did not manage to follow the logic of restricting the sum to special subpairs since it seems incompatible to the traditional KL definition

Why not optimize the KL difference between the signal predicted by the two theories in a more direct way? There may be a good reason for the extra complexity but it's not coming across from the text

given concerns about data limitations, is there a way to take into account co-occuring behavioral responses as a way to improve precision or at least to address model mismatch?

How does the number of neurons enter the information gap expressions ? It seems counterintuitive that one can achieve the assymptotic precision for representing a continuous variable with 10 neurons... does that depend on the precision of the x discretization in a substantial way?

---

> ### Author Response · Authors · 2025-11-21
> **Rebuttal by Authors**
>
> We thank the reviewer for the thorough and constructive review. We are encouraged by your **positive assessment** of our theoretical formulation is **“sensible and fruitful”**, and gives **“precise mathematical form,”** which is **novel (“no formal framing for such experimental design exists to date.”)**, **“generally useful”** and **“paves the way for a more precise experimental validation.”** Below we address your concerns point-by-point and describe improvements in the revision.
>
> ### Main Concerns
>
> > Q1: ... the writing overall is overall pretty clear, ... the information gap is difficult to parse and ambiguous. the logic of the construction is not clearly enough explained in that part of the text. The appendix explains it better ... but the main text cannot really be understood on its own
>
> Thank you for pointing this out. In the revision, we clarified the logic directly in Sec. 2 by clearly stating that the information gap is defined as the expected increase in cross-entropy incurred when a decoder is forced to extract mismatched probabilistic content, and explaining the derived formula (Eq. 1, 3). These clarifications make the main text more self-contained, while keeping full derivations in the appendix.
>
> > Q2: ... the results do not seem to provide much insight into how one could design better experiments beyond giving ranges of possible d' parameters that look good ... It is not clear if the (not negligible) experimental data collected in this setup is outside that regime so that experimentalists would know to do something differently.
>
> We would like to clarify that our framework provides **concrete, actionable guidance** for experiment design that goes beyond reporting parameter ranges. It introduces a **principled quantitative metric**—the information gap—that identifies optimal stimulus priors for distinguishing likelihood vs. posterior codes, and it enables experimenters to **directly assess whether their own tasks fall within an informative regime**.
> - **Concrete optimal parameters for task design**: The information-gap landscapes (Fig. 5) specify exact optimal task parameters for each contrast level (e.g., for low contrast stimuli, d≈30°, σ≈20°). These values are not heuristic: they correspond to the **maxima of $\Delta^{info}$** ​and closely match empirical decoder performance difference on simulated populations (Fig. 3–4). Thus, the framework provides specific prior distributions that experimenters should implement.
> - **Identification of unsuitable priors (“what not to do”)**: The framework also predicts when experiments will fail to distinguish coding hypotheses, even with large datasets. Heavy-tailed priors (Student-t, Cauchy) yield *near-zero* posterior information gaps (Fig. 6), making them ineffective for adjudicating coding hypotheses.
> - **Direct evaluation of an experimenter’s own design**: Because experimenters know their task parameters (d,σ), they can **directly compute the predicted $\Delta^{info}$** using Eq. 1 and 3. This immediately indicates whether their task design can differentiate the two hypotheses. For example, a design with d=10°, σ=30° would generate low information gap, and would be insufficient to distinguish coding hypotheses. This provides the diagnostic test enabling experimenters to **quantitatively assess and adjust their task design**.
>
> ### Specific Questions
>
> > Q3: How does the animal's own added sensory uncertainty get incorporated in the procedure ? can the (unavoidable) model mismatch bias the conclusions in any way ... ?
> & Q9: ... is there a way to take into account co-occuring behavioral responses as a way to improve precision or at least to address model mismatch?
>
> Although our main analysis focuses on the *theoretical decoding limit* using optimal priors, our framework naturally accommodates model mismatch by **incorporating behavioral (psychometric) measurements**. The procedure is now detailed in Appendix A.5 and Fig. 12 in the revision:
> - **Analyze the psychometric curve**: In perceptual tasks (e.g., orientation discrimination tasks), deviations in the subject’s psychometric curve (correct rate as a function of stimulus orientation) from the ideal observer reveal model mismatch or biased priors.
> - **Infer the subject’s biased prior**: Features (such as leftward shifts or increased slope/variance) in the psychometric function can be mapped to corresponding biases or increased uncertainty in the subject’s internal mismatched prior.
> - **Compute the information gap using the inferred prior**: The inferred biased prior can then be used directly in our information-gap calculation, yielding predictions that account for the subject’s model mismatch and more accurately reflect expected empirical decoder differences.

---

> > ### Author Response · Authors · 2025-11-21
> > **Rebuttal by Authors (cont.)**
> >
> > > Q4: ... what if the local neural population encodes beliefs about related random variables but not the exact parametrization of the experiment? ... neural sampling ... use a high dimensional distributed representation (think probabilistic versions of factor analysis or gaussian scale mixtures) related to orientation but not beliefs about theta per se
> >
> > Thank you for raising this important point. We agree that early sensory populations—especially under sampling-based models—may encode beliefs over high-dimensional latent variables rather than the experimenter-defined parameter $\theta$. Our framework does not require that the neural population explicitly represents $\theta$ *itself*. Instead, it only assumes that neural responses encode some probabilistic distribution over internal latent variables, and that $\theta$ is generated from these latents through the experimenter’s generative model.
> >
> > 1. Our framework is **agnostic to the dimensionality and parameterization of the latent code**: In the paper, we denote $\theta$ as the task-relevant latent variable, but the derivation of the information gap only requires a generative model of the form: latent state $z$ → sensory observation $x$ → neural responses $r$.
> > The latent state $z$ can be arbitrarily high-dimensional (e.g., GSM latent scales, multi-factor generative representations). The decoder’s objective is defined with respect to the task distribution over $\theta$, but $\theta$ may itself be a deterministic or stochastic function of the latent variables $z$ the cortex actually represents. In such cases, both likelihood- and posterior-based decoders simply learn the appropriate mapping between $r$ and the task variable $\theta$ without requiring that neural responses encode $\theta$ directly.
> > 2. **Posterior modulation remains the discriminative feature** even when the encoded variables are not $\theta$: Under posterior coding (including sampling-based codes operating over high-dimensional latents), context priors modulate the posterior over $z$ and thus the distribution over $r$. This is precisely the measurable difference our information gap quantifies. Thus, the predictions of our framework do not depend on whether V1 expresses $\theta$ or a higher dimensional distributed code—as long as priors modulate neural responses in posterior-coding models and not in likelihood-coding models.
> > 3. Empirical validity: **the decoder does not assume explicit $\theta$-coding**: Our decoders receive only population responses $r$ and are trained to decode probabilistic quantities. If V1 actually encodes samples of a high-dimensional latent vector $z$ rather than $\theta$, the decoder internally learns $p(\theta | r)$, which implicitly performs the composition $r$ → $z$ → $\theta$.
> >
> > In summary, our framework remains valid when the neural population encodes beliefs over high-dimensional latent variables rather than $\theta$ itself; the key discriminative prediction—whether context priors modulate $r$—still holds and is exactly what the information gap quantifies.
> >
> > > Q5: ... confused about terminology, e.g. "true posterior ...of a ideal likelihood decoder"? ... an optimal decoder of likelihood or posterior beliefs from neural responses
> >
> > Thank you for pointing out the confusion. We have now revised the writing in Sec. 2 to make this more clear.
> >
> > > Q6: ... the information gap seems to come from the ability of the agent to correctly know or infer contex ...
> > & Q7: Please explain the need for hypothesis specific information gaps, what assumptions it makes about context unknown r, and what it implies for experimental design: do you assume data exists for both a single/cued context version and an inferred/uncued context version of the same task? ...
> >
> > We would like to clarify that our proposed experimental paradigm **does not** require subjects to infer the context. Instead, contexts are **explicitly cued** (e.g., via visual or auditory signals, as in Qamar et al., 2013; Walker et al., 2020). This ensures that subjects adopt the intended correct context-specific prior $p^c​(\theta)$ rather than engaging in an additional inference process about the context itself. Consequently, the neural responses $r$ directly reflect the appropriate context prior under the posterior-coding hypothesis, and the information gap we derive isolates differences in sensory uncertainty representation, rather than differences arising from contextual inference.

---

> > > ### Author Response · Authors · 2025-11-21
> > > **Rebuttal by Authors (cont.)**
> > >
> > > Regarding the need for hypothesis-specific information gaps, our derivations assume that the *context $c$ is known to the subject*. Under this assumption, the likelihood- and posterior-coding hypotheses predict systematically different mappings from stimuli to neural responses. The information gap then quantifies the expected degradation in decoder performance when the decoded probabilistic quantity (likelihood vs. posterior) is mismatched with the true underlying neural code. Importantly, our framework does not assume the availability of both cued and uncued versions of the same task; all derivations in Appendix A.1 are derived under the cued-context paradigm described above.
> > >
> > > We have added these clarifications to Sec. 2 and Appendix A.1 to make the logic, assumptions, and experimental implications explicit.
> > >
> > > > Q8: Eq3: ... the logic of restricting the sum to special subpairs since it seems incompatible to the traditional KL definition. Why not optimize the KL difference between the signal predicted by the two theories ...
> > >
> > > We appreciate the opportunity to clarify the logic, as it is central to our derivation and importantly, the restriction is not an arbitrary modification of the KL definition, but the *direct outcome of the derivation* for the posterior-coding hypothesis.
> > >
> > > Under posterior coding, the likelihood decoder fails only when two different observations $(x_j,x_k)$ generate identical posterior populations across contexts, i.e., when $p^A(\theta | x_j) = p^B(\theta | x_k)$ (Eq. 4). In these cases, the decoder receives the same input but must output different likelihoods, which is impossible for any deterministic decoder; therefore these pairs contribute a non-zero KL term. Eq. 3 simply sums over the pairs that yield *non-zero KL divergence*.
> > >
> > > For all other observations not in the set of pairs of $(x_j,x_k)$, the decoder can be perfect, so the KL term is exactly zero. Intuitively, for cases where posteriors are unique to a single $x_i$, the likelihood decoder can achieve perfect decoding as there is no ambiguity, and the KL term is zero. As a result, these observations which do not satisfy Eq. 4 would yield a zero KL divergence and thus do not contribute to the information gap; therefore, the sum in Eq. 3 includes only those pairs with non-zero divergence. Our expression is not a restriction but reflects the end result of the derivation that identifies pairs which have non-zero KL divergence.
> > >
> > > > Q10: How does the number of neurons enter the information gap expressions? It seems counterintuitive that one can achieve the assymptotic precision for representing a continuous variable with 10 neurons... does that depend on the precision of the x discretization in a substantial way?
> > >
> > > We would like to clarify that the *analytical expression* of information gap does *not* depend on the number of neurons. This is because $\Delta^{info}$ quantifies the decodable information under ideal decoding directly. In contrast, in simulations (and real data), information must be encoded in finite neural population responses, so *empirical* decoder performance naturally depends on population size.
> > >
> > > To address this point more directly, we conducted additional ablations examining the factors that determine how quickly empirical performance converges to the theoretical information gap:
> > > 1. **Coverage of the orientation space**: Our main simulations assume that *neural tuning curves tile the full orientation space*, consistent with standard V1 models (e.g., Rubin et al., 2015). Without such coverage, no decoder can recover information about orientations lacking tuned neurons. In an ablation where tuning-curve centers were randomly sampled rather than tiled (Fig. 15A), we find that convergence with respect to neuron count becomes substantially slower—consistent with the reviewer’s intuition.
> > > 2. **Number of trials**: In Fig. 3, the neuron-scaling experiment uses 30k trials so that decoders approach the theoretical limit fast. In an additional ablation with fewer trials (3k, Fig. 15B), convergence is again slower because the decoder cannot reliably estimate the encoded distributions from limited data.
> > >
> > > In practice, the above two factors can be mitigated by modern population recordings that provide hundreds to thousands of simultaneously recorded neurons and by increasing trials.
> > >
> > > 3. Discretization of the orientation variable: We repeated the convergence analysis with higher-resolution orientation bins (0.25° instead of 1°; Fig. 15C), and obtained comparable results as before. This confirms that the theoretical $\Delta^{info}$ and its empirical convergence are **robust to discretization bin size** and reflect the underlying decodable information rather than numerical artifacts.
> > >
> > > We have added these analyses and clarified the methodology in Appendix A.8 and Fig. 15 in the revision to make explicit how neuron count, coverage, and sample size affect empirical convergence to the theoretical information gap.

---

### Official Review · Reviewer_EZQq · 2025-10-30

**Soundness:** 3
**Presentation:** 2
**Contribution:** 3
**Rating:** 6
**Confidence:** 4

**Summary:**

Different theories have been proposed for the neural representations of early visual areas during perceptual decision making tasks. Some proposed that neural population activity represents the likelihood function over the stimulus variable (likelihood theory), while others argued that neural activity represents samples from the posterior distribution of the stimulus variables (posterior theory). So far, it has been difficult to adjudicate between these theories, or falsify both. This theoretical paper proposed a information-theoretical approach toward better ways to distinguish these theories experimentally, although no experimental results were given or analyzed in this paper. Specially, the authors proposed a metric called information gap, that captures the performance difference between the likelihood decoder v.s. posterior decoder. They were able to analytically derive the information gap under certain conditions. Numerical experiments were used to validate these results. They further used this approach to analyze some simple perceptual decision making tasks and made suggestion about how to better design experiments to test the different theories mentioned above.

**Strengths:**

This paper seeks to address a problem that has been difficult to make progress on in the field.  The proposed approach is sound.

The proposed analytical approach and the results are new, although they only seem to work under some relatively simple task conditions.

The numerical simulations in the paper were thorough and served to validate the analytical results.

The approach may provide some useful guide for the design of future experiments to testing the likelihood theory v..s posterior theory of the neural responses (although there may be some questions of practically how to find the optimal design as the optimal design seems to depend on some parameters, e..g, noise in the system, that might not be easily measured in the experiments).

**Weaknesses:**

— While the authors clearly spent efforts to articulate the theoretical settings and to justify the importance of the problem, I felt there was still some ambiguity. For example, the two theories outlined were the theory of representing likelihood v..s sampling-based code. However, here sampling seems to be an additional assumption. It would be cleaner if the comparison was simply a presentation of likelihood v.s. a representation the posterior. One can also consider a theory of sampling from the normalized likelihood (which would be equivalent to posterior assuming a uniform prior). It was unclear wether sampling is essential in the model comparison.

— the paper only consider prior imposed during the task. It seemed that the results were based on the assumption that the task prior can be learned perfectly. Yet, this assumption may be too strong.

-- What about the long-term priors (or natural priors) the observers already learn? Can the theoretical framework also apply to these? Or would these complicate things?


-- My most substantial concern is that the authors only considered a specific path for a relatively narrow setting (although under these settings, the results seem to be solid). I am not sure if the approach would indeed be fruitful or whether there might be easier and more fruitful approach for adjudicating between the different theories. In particular, from Section 4, I am under the impression that practically it would still be difficult to distinguish these theories (Correct me if I have the wrong impression).  I wonder if there would be more fruitful path that would also be easier. For example, instead of using a static setting, how about considering a learning paradigm instead? Suppose we start with a baseline condition, and measure the neural responses. Then we teach the observers to learn a different prior. Under the likelihood theory,, the neural representation in V1 should not change. In contrast, based on the posterior theory, the neural representation in V1 should change according to the prior learned.  So the question is, what is the advantage of the approach proposed in the current paper compared to such an alternative approach?

-- The paper would be stronger if some empirical results from experiments can be shown. Even a re-analysis some previously reported experimental data to demonstrate the advantage of the current method would be useful.

**Questions:**

In fig. 2c, top row, why would one still incorporate the priors given it is a likelihood decoder? Also, were ground truth prior assumed there?

Line 249, what does the “task-marginalized” estimator mean? Please unpack the idea.

What was the ground truth posterior/likelihood in the simulation studies? Were neural noise considered in addition to the stimulus noise?

From Fig. 6, it appears that the optimal experimental design depends on the noise in the system. Practically how can we determine the noise level before designing the experiments to test the theories?

Fig. 6, what are the units on the x- and y-axis?

The authors show the using heavy-tailed priors would not help better distinguish different models. Would using distribution with thinner tail than Gaussian help? What about uniform priors?

---

> ### Author Response · Authors · 2025-11-21
> **Rebuttal by Authors**
>
> We thank the reviewer for your thorough and constructive review. We are encouraged that you found our work provides a **“sound approach”** to **“address a problem that has been difficult to make progress on in the field"** with **"new analytical results”** and **"thorough numerical validation”**. We appreciate your **positive assessment** that our approach can **provide useful guidance for the design of future experiments**. Below we address each question point-by-point and report additional analyses.
>
> ### Main Concerns
>
> > Q1: ... However, here sampling seems to be an additional assumption. ... if the comparison was simply a presentation of likelihood v.s. a representation the posterior ... also consider a theory of sampling from the normalized likelihood ... It was unclear whether sampling is essential ...
>
> We agree completely, and would like to clarify, that our comparison is formulated as likelihood representation vs. posterior representation, rather than specific implementation mechanism. Focusing on what probabilistic quantity is represented, our framework provides a concrete first step towards differentiating different hypotheses without getting muddled with the nuanced details of model formulation in specific theories.
>
> Key clarification: While we use neural sampling codes as one example realization of posterior coding, **sampling is an orthogonal question to our framework**. Our information gap applies to *any* hypothesis where sensory populations represent the posterior, regardless of whether this is achieved through sampling or other mechanisms.
>
> As the reviewer notes, one could also consider sampling from normalized likelihood (equivalent to posterior with uniform prior). Our framework naturally accommodates this: such a code would be classified as likelihood coding, since it wouldn't be modulated by context-specific priors. This highlights how our framework focuses on the **functional/ representational distinction** rather than implementation details.
>
> We incorporated the clarification in the revision (Sec. 1) to specify that neural sampling is only used as an illustrative example of posterior coding.
>
> > Q2: ... based on the assumption that the task prior can be learned perfectly. Yet, this assumption may be too strong.
>
> Thanks for raising this question. Following the literature on perceptual decision-making, while we agree that the assumption of perfect prior learning is strong, we’d like to clarify it for several reasons:
> 1. **Behavioral evidence supports near-optimal Bayesian inference**: Extensive psychophysical studies demonstrate that well-trained humans and animals learn and adjust to task priors.
> 2. **Ideal observer framework provides the theoretical foundation**: Our approach follows the widely-adopted ideal observer framework used in studying perceptual decision-making under uncertainty (Ma et al. 2006; Beck et al. 2008; Walker et al. 2020), providing a foundation upon which more complexity could be incorporated.
> 3. **Our framework could incorporate imperfect prior given empirical psychophysical results**: While our theoretical analysis evaluates the optimal-decoding limit, the same procedure can incorporate subjects’ internal priors by grounding the analysis in behavior measurements. Imperfect/ biased prior of a subject could be inferred from their psychophysical curve deviation from an ideal one, which could be further used to modify the information gap calculation by using the biased prior instead of the perfect prior. We detailed this procedure in Appendix A.5 and Fig. 12 in the revision.
>
> > Q3: What about the long-term priors (or natural priors) the observers already learn? Can the theoretical framework also apply to these? ...
>
> Our focus on manipulable task priors is strategic: by experimentally varying priors across contexts, we create the differential signal needed to distinguish the two coding hypotheses. Long-term/ natural priors would act as a **constant baseline** to both coding hypotheses equally, leaving the distinguishing information gap unchanged. If long-term/ natural priors are known (e.g. estimating from the environmental statistics), they could be incorporated into our framework as a fixed component, but experimental manipulation of task-specific priors remains the essential component for generating discriminative power.

---

> > ### Author Response · Authors · 2025-11-21
> > **Rebuttal by Authors (cont.)**
> >
> > > Q4: ... still be difficult to distinguish these theories (Correct me if I have the wrong impression). ... how about considering a learning paradigm instead? Suppose we start with a baseline condition ... Then we teach the observers to learn a different prior. ... what is the advantage of the approach proposed in the current paper compared to such an alternative approach?
> >
> > We appreciate the reviewer’s thoughtful suggestion. A learning-based paradigm is indeed an interesting complementary direction, especially because it could reveal the *trajectory* by which population responses evolve as observers acquire new priors. However, we see our context-switching design as offering a more experimentally practical and analytically stable setting for adjudicating the two probabilistic coding hypotheses.
> >
> > First, a learning paradigm introduces additional complexities—longitudinal plasticity, attentional changes, and adaptation—that can affect V1 responses independently of prior-dependent probabilistic content, making interpretation difficult. In contrast, studying animals at “steady state,” after they have fully learned two context-specific priors, provides a **cleaner window into the encoded information**.
> >
> > Second, learning experiments would demand long-term, highly stable population recordings as the prior is acquired—an ongoing technical challenge despite rapid progress in both animal training and chronic recording methodologies. By contrast, context-switching allows for session-based comparisons from the same neural population, which is essential for applying our decoding-based assessment in a well-controlled manner.
> >
> > Finally, our information-theoretic framework is not tied to static designs: it could also be applied to optimize stimulus selection within a learning paradigm itself. We view such experiments as an exciting future extension, but believe that context switching offers a more tractable and reliable starting point for decisively distinguishing likelihood versus posterior coding.
> >
> > > Q5: ... some empirical results from experiments can be shown. Even a re-analysis some previously reported experimental data to demonstrate the advantage of the current method would be useful.
> >
> > Thank you for the suggestion. We fully agree that empirical validation is important. Below we provide additional analysis on existing datasets, which illustrates why the existing dataset fails at adjudicating between coding hypotheses and underscores the application of our framework for designing more decisive experiments.
> >
> > - **Re-analysis of existing datasets confirms the limitation of previous experimental designs**: We conducted additional analysis on the Allen Visual Coding dataset, a publicly available large-scale neural recording dataset from V1 population. However, conventional experimental designs—including those in Allen Visual Coding and Walker et al. 2020—use only a *single stimulus prior* throughout recording sessions. Under such single-context experiments, our theory **predicts zero information gap ( $\Delta^{info}=0$)**, and our **analysis of the Allen dataset shows that likelihood and posterior decoders achieve indistinguishable performance (Fig. 7, performance difference=0.0024$\pm$0.064, p=0.63).** This demonstrated that existing experimental data are not suitable because they lack the critical experimental manipulation we propose: the same neural population responding under different context priors.
> >
> > - **Guiding future experiments**: This analysis demonstrates that our framework provides principled guidance for designing new experiments with the multi-context manipulation necessary to differentiate probabilistic coding hypotheses. The information gap landscapes (Figs. 5-6) offer *concrete, actionable experimental parameters* that we hope will guide future empirical studies.
> >
> > We included this empirical analysis in the revision (Sec. 5 and Fig. 7) and emphasize that our contribution on providing theory-driven experimental design for future targeted experiments.
> >
> > ### Specific Questions
> >
> > > Q6: In fig. 2c, top row, why would one still incorporate the priors given it is a likelihood decoder? ...
> >
> > The ground truth context priors $p^c(\theta)$ are provided to the likelihood decoder because this is necessary to convert the decoded likelihood functions into posterior distributions for cross-entropy minimization, as demonstrated by previous DNN-based decoders where the ground truth priors are provided following an ideal-observer approach (Walker et al. 2020). This design allows the DNN decoder output to converge to the likelihood function, reflecting how the likelihood information from the population should be incorporated with the prior. We have detailed this architecture in Appendix A.3.2.

---

> > > ### Author Response · Authors · 2025-11-21
> > > **Rebuttal by Authors (cont.)**
> > >
> > > > Q7: Line 249, what does the “task-marginalized” estimator mean? Please unpack the idea.
> > >
> > > We appreciate the opportunity to clarify this. The "task-marginalized" estimator refers to a posterior distribution estimator that a decoder will converge to when decoding mismatched quantities (e.g. decoding posterior from a likelihood-coding population, or decoding likelihood from a posterior-coding population), which is derived by marginalizing over the task context distributions.
> > >
> > > For likelihood-coding populations, since context information is not encoded, the best possible posterior decoder can only converge to a context-averaged posterior—it cannot determine which specific context generated each trial. This yields the surrogate posterior given in Eq. 2, which marginalizes priors across contexts.
> > >
> > > For posterior-coding populations, the best possible likelihood decoder marginalizes over contexts to handle ambiguous mappings from identical neural responses to different likelihoods, leading to the likelihood estimator as defined in Eq. 5.
> > >
> > > We have expanded the explanation in the main text (Sec. 2) for better clarity, and the detailed derivation is provided in Appendix A.1.
> > >
> > > > Q8: ... the ground truth posterior/likelihood in the simulation studies? Were neural noise considered in addition to the stimulus noise?
> > >
> > > In the simulation studies, the ground truth likelihood is modeled as Gaussian distributions to reflect Gaussian orientation tuning curves commonly found among simple V1 neurons. The ground truth posterior is computed via Bayes' rule (multiplying the likelihood with the prior, with normalization). We added an additional Fig. 18 to give an example of the ground truth likelihood and posteriors, and the detail of simulation setup is in A.3.1.
> > >
> > > Neural noise is implemented through Poisson variability in spike counts, which is added on top of stimulus noise (captured by the generative model). We additionally implemented a more complex, gain-modulated Poisson neuron model for simulating neural noise. The details are summarized in A.3.1.
> > >
> > > > Q9: From Fig. 6, ... the optimal experimental design depends on the noise in the system. Practically how can we determine the noise level ...?
> > >
> > > In practice, the relevant noise level can be estimated before running the main experiment using standard and fast procedures:
> > > - *Pre-experiment neural noise measurement*: Record population responses to the stimulus to **estimate tuning curves** and **trial-to-trial variability (e.g., Fano factors)**. These measurements provide empirical noise estimates that directly inform the information-gap calculation.
> > > - *Behavioral calibration*: A short orientation-discrimination block yields a **psychometric function**, providing an estimate of effective sensory noise and biased prior from behavior.
> > > - *Iterative refinement*: Because computing the information-gap landscape is inexpensive, researchers can update their noise estimates and re-optimize task parameters after a brief pilot phase.
> > >
> > > Finally, increasingly available large-scale neural recordings, as well as DNN-based models of sensory populations, provide additional ways to *estimate realistic neural noise levels* prior to experiment design. In sum, the noise levels required for our optimization can be readily estimated with lightweight calibration steps, making the proposed design procedure practical and easy to implement.
> > >
> > > > Q10: Fig. 6, what are the units on the x- and y-axis?
> > >
> > > Thanks for pointing this out. Both the x- and y-axes units are measured in degrees of orientation (°). We have added the units to all related figures in the revision.
> > >
> > > > Q11: ... heavy-tailed priors would not help better distinguish different models. Would using distribution with thinner tail than Gaussian help? What about uniform priors?
> > >
> > > Would thin-than-Gaussian priors help? No. We reported additional analyses using thin-tailed distributions (generalized normal distributions with $\beta>2$) in Fig. 16. The information gap landscape indicates that thin-tailed priors similarly lead to near-0 posterior-coding information gap across task parameter space. Our framework provides a similar explanation: the resulting posteriors under thin-tailed priors become highly asymmetric across contexts (Fig. 17), reducing the feasible set of observation pairs that can satisfy Eq. 4, thereby shrinking the posterior-coding information gap—mirroring the failure mode observed with heavy-tailed priors.
> > >
> > > What about uniform priors? No. As shown in Fig. 17, a uniform prior induces no context-dependent modulation of the posterior. Hence, likelihood- and posterior-coding populations become nearly indistinguishable, causing the information gap to collapse for both hypotheses.
> > >
> > > We added these analyses in Fig.16, 17 and discussed in Appendix A.9 in the revision.

---

### Official Review · Reviewer_1gnw · 2025-11-01

**Soundness:** 3
**Presentation:** 3
**Contribution:** 2
**Rating:** 2
**Confidence:** 4

**Summary:**

The authors develop a framework for designing a pair of stimulus ensembles that are optimized to differentiate two models of probabilistic sensory representation for Bayes-optimal observers: the "likelihood" model, and the "posterior" model.  They show through a number of simulations that the method can successfully distinguish the two models.

**Strengths:**

* Optimal experimental design is an important and under-studied area.

* The problem posed, of developing an analysis to compare two standard models proposed for Bayesian inference in the brain, and using this to derive optimal stimulus ensembles for experimental determination of a "winner", is relevant and I'm not aware of any published solutions.

**Weaknesses:**

* The development is, in my view, over-formalized given the intended use and implementation.  Why is it not sufficient to seek two stimulus ensembles that produce the largest discrepancy in estimates produced by the two models?  And how do the results depend on the design and training of the deep neural network decoders that are used to obtain likeilihoods or posteriors from the neural population?

* The paper is built on the direct comparison of two models (likelihood encoding vs. posterior encoding), but there are other proposals in the literature as to how prior probabilities might be embedded in neural systems.  In particular, Ganguli & Simoncelli (2010/2014) propose that inhomogeneous populations of tuning curves encodes the prior, the spiking responses form a likelihood, and a weighted nonlinear decoder can approximate Bayesian estimates.  There's reasonable experimental support for these components, but it's not clear how this formulation would be categorized in the context of the authors' proposed experimental test.

* The authors' formulation relies on a highly simplified description of how an organism (human or otherwise) adapts to a sensory environment. In particular, they assume the posterior observer is aware of the true prior, and is able to use it properly for inference, which seems inconsistent with the large literature on adaptation.  Perceptually, adaptation to an ensemble leads to changes in perceived value (biases) as well as changes in discriminability (variance), but the changes reported in the literature are diverse (especially changes in discriminability), and highly dependent on the perceptual attribute being tested, the adaptation stimuli, the duration of exposure, etc.  In fact,  perceptual adaptation effects have been described as "anti-Bayesian", because adapting to one stimulus (or a distribution concentrated in one region of the the stimulus space) leads to perceptual repulsion rather than attraction toward this modified "prior". Physiologically, the most commonly observed effect of adaptation is a reduction in gain, but many reports describe shifts or distortions (e.g., flank suppression) of tuning curves.  Again, these effects are quite variable, and depend on details of adapting stimulus ensemble, duration, etc.   Recent work by Tring et. al. (2024) provides a compelling explanation for observed adaptation effects in terms of coding efficiency, but this seems at odds with either the "likelihood"  or "posterior" hypothesis of the current paper.  So my concern is that experimental design proposed by the authors will lead to inconclusive results, because adaptive changes in perceptual systems are substantial (so not aligned with the "likeilihood" model), but are also inconsistent with a change in prior.

* In describing prior literature, the paper should make mention of work on optimal stimulus design for estimating perceptual discrimination thresholds [eg: Quest (watson/pelli 1983), Pest (Medigan 1987), Information-theoretic stimulus optimization for neurophysiology (Lewi etal 2006/2009), Quest+ (Watson 2017)]. Perhaps more relevant, work by Machens et al 2005 adaptively sought a stimulus ensemble for which a neural population (grasshopper auditory receptors) provided an optimally efficient code.  Not the same as Bayesian decoding, but analogous in spirit.

**Questions:**

See above.

---

> ### Author Response · Authors · 2025-11-21
> **Rebuttal by Authors**
>
> We thank the reviewer for the constructive review. We are encouraged that the reviewer recognizes our work addresses the questions of **“optimal experimental design is an important and under-studied area”** and tackles a **"relevant" problem** in a **novel** way for which they are **"not aware of any published solutions."** We appreciate the **positive assessment of our soundness and presentation**. Below we address the main concerns point-by-point and report additional analyses that will help clarify our contribution.
>
> ### Main Concerns
>
> > Q1: ... over-formalized ... Why is it not sufficient to seek two stimulus ensembles that produce the largest discrepancy in estimates produced by the two models? ... depend on the design and training of the deep neural network decoders ...
>
> We appreciate the question and clarify the contribution of our theoretical framework.
> 1. Why a formal framework is needed beyond “finding two ensembles with maximal discrepancy.”
>
> While the intuition of “seek the largest discrepancy” is correct, our framework provides the principled way to identify such designs across a high-dimensional task design space.
> - **Intuition alone cannot resolve the core design tradeoff**: As shown in Fig. 7 and discussed in lines 161–167, the priors must (i) *differ enough* to drive divergent predictions under the two hypotheses, yet (ii) retain *sufficient overlap* to compare neural responses to identical stimuli across contexts—the key stimuli needed to distinguish likelihood vs. posterior coding. Naïvely maximizing prior separation collapses this overlap and eliminates distinguishability for posterior coding hypothesis. This fundamental tradeoff cannot be reliably solved through intuition or heuristic search.
> - **Information-gap derivation reveals where distinguishability arises**: For likelihood coding, all observations contribute to the KL gap; for posterior coding, only pairs satisfying Eq. 4 (identical posteriors but distinct likelihoods) matter. This structure is not discoverable through heuristic stimulus search.
> - **Optimal parameters vary non-trivially**: As shown in Fig. 5, the optimal stimulus priors depend on contrast level and differ across hypotheses in ways that are neither monotonic nor intuitive. These non-trivial patterns could not be discovered without the analytical framework.
>
> As a result, the formal development is not an embellishment; it exposes the precise conditions under which the hypotheses are theoretically distinguishable and produces task designs that heuristic stimulus search would miss.
>
> 2. Dependence on the design or training of deep neural network decoders.
>
> Our results—and the task optimization—do **not** depend on specific decoder architectures or training idiosyncrasies:
> - The **theoretical information gap is derived using Bayes-optimal decoders** in the limit of perfect decoding, independent of any neural network.
> - Neural networks are used **only to empirically verify** that practical decoders converge to these theoretical limits (Fig. 3–4).
> - The optimized task designs are therefore **decoder-agnostic predictions** for experiments; the deep networks serve only as a validation tool, not as part of the theoretical framework.
>
> In short, the formal derivation is necessary to characterize the structure of the distinguishability problem and to produce decoder-independent, theoretically grounded task designs. The neural network decoders further confirm the theoretical prediction.
>
> > Q2: ... other proposals ... as to how prior probabilities might be embedded in neural systems. ... Ganguli & Simoncelli (2010/2014) ... not clear how this formulation would be categorized in the context of the authors' proposed experimental test.
>
> The Ganguli & Simoncelli (2010/2014) proposal combines heterogeneous tuning curves—which embed the prior—with spiking variability that reflects the likelihood, yielding a hybrid code in which sensory responses carry both likelihood information and a structurally instantiated prior. This example can be categorized as a **mixed or intermediate hypothesis**, in between the canonical pure likelihood and pure posterior coding hypotheses.
>
> *How this model fits into our experimental test*: Our framework can naturally accommodate such hybrid hypotheses by evaluating how each decoder performs under mismatched information. Under the Ganguli–Simoncelli model, both the likelihood and posterior decoders can recover the correct distributions, yielding an **information gap of $\Delta^{info}=0$**. This zero-gap signature is distinct and does not arise under optimized task designs for either pure likelihood- or pure posterior-coding populations, which produce reliably nonzero and separable values.

---

> > ### Author Response · Authors · 2025-11-21
> > **Rebuttal by Authors (cont.)**
> >
> > *Why task optimization remains critical*: Optimizing the task to maximally separate the two canonical hypotheses simultaneously **maximizes sensitivity to departures** from them. A mixed code that yields $\Delta^{info}=0$ under the same optimized design becomes cleanly identifiable as neither pure likelihood nor pure posterior. Thus the Ganguli–Simoncelli model illustrates how our method generalizes beyond the two extremes and provides a principled tool for distinguishing both pure and mixed coding hypotheses.
> >
> > *Scope of our theoretical focus*: More broadly, we do not claim that likelihood and posterior coding are the only relevant theories, but they represent the two major families that differ in *what* probabilistic quantity is encoded. Our contribution is to provide a principled methodology for experimentally distinguishing such theories. By optimizing task parameters to maximally separate these canonical extremes, we simultaneously maximize sensitivity to discriminating more nuanced probabilistic coding theories.
> >
> > We have added discussion in Appendix A.6 and Fig. 13 in the revision to describe how our framework extends to the mixed coding hypotheses.
> >
> > > Q3: ​​... how an organism adapts to a sensory environment ... assume the posterior observer is aware of the true prior ... inconsistent with the large literature on adaptation. ... changes in perceived value (biases) as well as changes in discriminability (variance) ... perceptual adaptation effects have been described as "anti-Bayesian",...  Physiologically, the most commonly observed effect of adaptation is a reduction in gain, ...  Recent work by Tring et. al. (2024) ... in terms of coding efficiency ... adaptive changes in perceptual systems are substantial (so not aligned with the "likeilihood" model), but are also inconsistent with a change in prior.
> >
> > Thank you for this thoughtful question. We appreciate raising adaptation as an important consideration. Clarifications will help avoid conflating this effect which is orthogonal to our focus.
> >
> > 1. **Adaptation vs. our target: steady-state, well-trained encoding**: Our framework and proposed experiments target the well-trained steady-state of neural coding: we probe how sensory populations represent probabilistic information after subjects are *familiarized with both contexts* and *performance has stabilized*. The core question is therefore the **format of the steady-state probabilistic code** (does the population encode the likelihood or the posterior), not the transient dynamics of adaptation during exposure.
> > 2. **Experimental controls to avoid transient adaptation confounds**: To ensure we measure steady-state encoding rather than transient adaptation effects, our experimental protocol specifies the following practical steps:
> > - Familiarization: subjects/animals are trained on both contexts until behavioral performance and psychometric measures stabilize.
> > - Discard transients: early trials after context switches or session start are excluded from the analyses to avoid adaptation effects.
> > - Context presentation: contexts remain well-learned and distinct, and analyses focus on trials during steady performance. These choices isolate the question of *what information is encoded* once the subject has settled into a stable encoding regime.
> >
> > 3. Why this distinction matters but does not undermine our framework: Our information-theoretic metric evaluates the *information present in steady-state population responses* under the two coding hypotheses. This is appropriate for distinguishing coding *format* because it characterizes the asymptotic representational consequences of each hypothesis. Moreover, adaptation dynamics are a separate phenomenon: (a) transient adaptation can be excluded by the protocol above, and (b) steady-state changes are precisely the regime in which one can test whether priors modulate early sensory responses in a manner consistent with posterior coding.
> >
> > 4. **Incorporate suboptimality**: Although our theoretical analysis derives ideal decoding limits, the framework naturally accommodates suboptimal or biased internal priors:
> > - Measure psychometric functions during the steady-state, and use deviations (shifts, slope changes) to infer internal priors.
> > - Replace the ideal prior with the inferred prior in the information-gap calculation to produce subject-specific predictions that account for model mismatch.
> > This makes the approach robust to realistic behavioral biases.
> >
> > 5. Relation to coding-efficiency accounts: Coding-efficiency explanations (e.g., Tring et al. 2024) describe mechanisms that can modify representations. Our work addresses the complementary question of what probabilistic information is contained in steady-state representations.
> >
> > We have updated Section 2 in the revision to explicitly state that we probe well-trained phases of neural encoding, and explain how to incorporate internal priors from psychophysics in Appendix A.5 and Fig. 12.

---

> > > ### Author Response · Authors · 2025-11-21
> > > **Rebuttal by Authors (cont.)**
> > >
> > > > Q4: In describing prior literature, the paper should make mention of work on optimal stimulus design for estimating perceptual discrimination thresholds [eg: Quest (watson/pelli 1983), Pest (Medigan 1987), Information-theoretic stimulus optimization for neurophysiology (Lewi etal 2006/2009), Quest+ (Watson 2017)]. Perhaps more relevant, work by Machens et al 2005 adaptively sought a stimulus ensemble for which a neural population ...
> > >
> > > We sincerely thank the reviewer for suggesting these relevant references and have incorporated them in Sec. 1 in the revision.

---

### Official Review · Reviewer_UTZ6 · 2025-11-04

**Soundness:** 2
**Presentation:** 3
**Contribution:** 1
**Rating:** 4
**Confidence:** 4

**Summary:**

This paper suggests a framework, i.e. testing KL Divergence of different decoders, to determine which hypothesis is true about the perceptual coding in the brain, likelihood or posterior. Tested on simulated data on gratings with different orientations and contrast levels, it illustrates the potential use of the method for real experiments.

**Strengths:**

The paper is well written, specifying the problem, process, and limitations clearly. It also gives a good coverage of the literature.

**Weaknesses:**

Using KL-divergence and decoding performance is pretty common in modelling in neuroscience, so I don't think using it for a specific problem could be considered a huge contribution. Furthermore, the paper lacks empirical results. There are multiple public data sets that actually contain gratings with different orientation and contrasts, for example Allen Brain observatory visual coding Neuropixel data set( https://portal.brain-map.org/circuits-behavior/visual-coding-neuropixels). So I think testing the practicality of the approach is feasible for the authors. While it is kind of acknowledged in the paper, the method relies on a lot of assumptions, and most importantly optimality of the neual code. There are multiple studies showing that the code is not perfectly optimal, including those referenced in the paper, e.g Haefner, et al, Neuron 2016 in which they explained a Bayesian approach that models the bias toward early observations. Overall, many studies on perception has shown that the neural activity is too complicated to be easily interpreted by models (and that's why we don't know what exactly going on as the authors mentioned too). I can't see how this approach could give a practical solution to that.

**Questions:**

- Do the simulation results hold when tested on the actual neural data (e.g on Allen data set mentioned above)?
- How does the firing rate and noise affect the results?
- How does the approach handle arousal, running, or other phenomena that affect the neural code in the visual cortex?
- Decoding approaches especially by powerful methods like deep neural networks are not trustable to many neuroscientists, as the presence of information is different from using the information. For example, one might give all the neural activity in the retina to a deep neural network and infer high-order features such as objects, just like how they can do it on raw images directly. This does not mean that the retina perform object recognition. how does this affect the feasibility of the presented approach?

---

> ### Author Response · Authors · 2025-11-21
> **Rebuttal by Authors**
>
> We thank the reviewer for their evaluation and for acknowledging our **“well-written” clear presentation and literature coverage**. We are encouraged that you found our work demonstrates **“potential use of the method for real experiments.”** We appreciate the opportunity to clarify several key aspects of our contribution. Below we address each concern point-by-point.
>
> ### Main Concerns
> > Q1: Using KL-divergence and decoding performance is pretty common in modeling in neuroscience....
>
> Thanks for the question, and we would like to take this opportunity to clarify our contributions. We agree that decoding analyses are widely used tools in neuroscience, and we clarify that our contribution is not the use of KL divergence itself. Rather, our key contribution is deriving a **new analytical metric**—the information gap—and a theoretical framework that makes the discriminability between coding hypotheses *tractable and actionable* for experimental design, which goes beyond prior work.
>
> Our framework provides, to our knowledge, the first *closed-form expressions* for hypotheses discriminability for given task designs. This establishes a principled bridge between a theoretically grounded metric and practical experimental design. Concretely, our contributions are:
> - **New information-theoretic metric for discriminability**: Our framework does *not* assume KL divergence as a chosen metric. Instead, starting from decoding mismatched probabilistic information, we *derive* that the expected performance difference admits a closed-form expression in terms of KL divergence (Eqs. 1 and 3). Our derivation reveals the metric governing discriminability between likelihood- and posterior-coding hypotheses. Prior decoding studies have not provided such an analytical link, nor a way to compute this discriminability directly or use it to guide experimental design.
> - **Principled experimental design**: Our framework is the first to show how to *optimize stimulus prior distributions* to maximize discriminability between two hypotheses. This moves beyond heuristic task design and provides a mathematically grounded recipe for selecting stimulus priors that most effectively adjudicate between different theories.
>
> > Q2: ... lacks empirical results. ... public data sets that actually contain gratings with different orientation..., for example Allen Brain observatory visual coding Neuropixel data set...
>
> We agree that empirical validation on real neural data is important, and appreciate the suggestion of Allen dataset. Below we analyzed and showed that existing public datasets like Allen Brain Visual Coding Neuropixels dataset are not suitable for adjudicating between coding hypotheses. The key insight is that conventional experiments lack recordings across different contexts, which allows  animals to adapt to different priors. The multi-context design is essential because the distinguishability between likelihood- and posterior-coding hypothesis arises precisely from how population responses change across different context priors.
>
> In contrast, existing datasets provide only a *single stimulus context* throughout recording sessions. Although they include a variety of stimuli (e.g., gratings of different orientations and contrasts), they do **not** manipulate the priors. To demonstrate this, we conducted additional analysis on the Allen Visual Coding dataset. Under such single-context experiments, our theory predicts **zero information gap ($\Delta^{info}$ = 0)**, and our analysis of the Allen dataset shows that **likelihood and posterior decoders achieve indistinguishable performance (Fig. 7, performance difference=0.0024$\pm$0.064, p=0.63).** Thus, these datasets cannot decisively differentiate the two coding hypotheses.
>
> This empirical result highlight why future experiments specifically designed with multi-context prior manipulations are necessary, and underscores that the theory-driven experimental design our framework provides could specify the conditions under which hypotheses can be differentiated. Our revision now includes the Allen dataset analysis (Sec. 5 and Fig. 7).

---

> > ### Author Response · Authors · 2025-11-21
> > **Rebuttal by Authors (cont.)**
> >
> > > Q3: ...optimality of the neual code... the code is not perfectly optimal... e.g Haefner, et al, Neuron 2016 in which they explained a Bayesian approach that models the bias toward early observations ...
> >
> > Thanks for raising this important point on the complexity of the neural code. We agree that real neural populations often deviate from Bayesian optimality, and we see this not as a limitation but as an opportunity for our framework to extend to settings with suboptimal neural codes or mismatched priors.
> >
> > While our theoretical analysis evaluates the optimal-decoding limit, the same procedure can **incorporate subjects’ internal priors**—biases, distortions, and suboptimalities included—by grounding the analysis in behavior measurements:
> > - **Estimate biases from psychophysics**: Deviations in the psychometric curve (e.g., shifts, slope changes) can be measured from subject’s behavior data.
> > - **Infer the subject’s internal prior**: These psychometric features can be used to infer internal priors that capture the subject’s suboptimal bias.
> > - **Compute information gap with the inferred prior**: Replacing the optimal prior with this inferred, subject-specific prior yields information-gap predictions that explicitly account for suboptimal processing.
> >
> > This extension preserves the practical applicability of our framework even when neural activity is not perfectly optimal, including the types of biases described in Haefner et al. (2016) as the bias towards earlier observations reflects how the animal interpreted and emphasized earlier observations over later ones. Instead of assuming optimality, we can quantify and incorporate the deviations.
> >
> > We have detailed this procedure in Appendix A.5. and Fig. 12 in the revision.
> >
> >
> > ### Specific Questions
> > > Q4: How does the firing rate and noise affect the results?
> >
> > In Fig. 4, we reported results on both standard Poisson and gain-modulated Poisson noise models—showing that despite variations in the neural noise model, the empirical decoder performance differences still tightly align with the theoretical values of information gap.
> >
> > Regarding firing rate and noise levels specifically, we conducted additional analyses (Appendix A.7 and Fig. 14 in the revision) showing that decreasing firing rates or increasing noise slows the convergence of empirical decoder performance differences. However, with sufficient data, the decoder differences converge to the same theoretical value. This reflects the expected effect where reduced signal-to-noise ratio might increase data requirements, but the information gap remains accurate predictors of the asymptotic decoder performance difference.
> >
> > > Q5: ... handle arousal, running, or other phenomena that affect the neural code in the visual cortex?
> >
> > The modulatory effect of arousal, running, and behavioral-states are largely orthogonal to the representational question our work addresses. They primarily modulate response gain or variability rather than the representational format—i.e., they do not determine whether a population encodes likelihood or posterior. Their influence typically appears as *stimulus-conditioned neural variability*. In practice, experiments can accommodate these influences in several ways:
> > - Control for behavioral state, which is standard practice in electrophysiology experiments
> > - Incorporate state-dependent gain or variance parameters into the generative model
> > - Analyze state-specific responses separately when state fluctuations are large or experimentally relevant
> >
> > Importantly, these modulatory effects do not invalidate the approach: just as neuroscientists routinely assess whether V1 encodes orientation despite attentional or arousal-related modulation, we can assess probabilistic coding format despite state-dependent modulatory changes. The distinction between likelihood and posterior coding remains well-defined under such nuisance variability.

---

> > > ### Author Response · Authors · 2025-11-21
> > > **Rebuttal by Authors (cont.)**
> > >
> > > > Q6: ... the presence of information is different from using the information. ... all the neural activity in the retina to a deep neural network ... does not mean that the retina perform object recognition ...
> > >
> > > We fully agree that successful decoding does *not in itself* imply that the brain *uses* the decoded information. This caveat applies to any decoding-based approach, and any decoded signal must ultimately be followed by targeted experiments assessing its behavioral relevance. In our framework, the decoders serve only as probes of what information must be present in the neural responses under each coding hypothesis, not as models of brain computation.
> > >
> > > The “retina object-recognition” example highlights a concern where a population from which an arbitrarily powerful decoder might extract information that the circuit does not compute. In contrast, the key tests in our approach rely on **context-dependent modulation** that cannot be manufactured purely by decoder:
> > > - Under likelihood coding, responses to identical stimuli should be invariant across contexts.
> > > - Under posterior coding, identical stimuli should evoke systematic context-dependent shifts due to prior modulation.
> > >
> > > These qualitative invariances are properties of the *neural representation itself*, not of the decoder. Our decoders simply measure the resulting differences in *decodable probabilistic content* that follow necessarily from the two hypotheses. Because our framework enables designing stimuli to maximally separate these predictions, the two theories already imply quantitatively different patterns of decodable information before interpretation.
> > >
> > > Thus, our method can be viewed as complementary to studies of functional relevance. It offers a principled way to adjudicate between two widely discussed theories—likelihood vs. posterior coding—by testing representational predictions that cannot be explained by decoder flexibility alone. Our framework provides a principled first step toward understanding how sensory uncertainty information may be represented and processed.

---

### Author Response · Authors · 2025-11-21
**Summary of rebuttal and revision**

We thank all reviewers for their thoughtful feedback and discussion, and for recognizing several strengths of our work, including:
- **Importance of optimal experimental design** for distinguishing probabilistic neural codes
- **Relevance of directly comparing likelihood vs. posterior theories**, a long-standing open challenge with no established solution
- **Soundness of our theoretical framework**, the novel analytical results, and the thoroughness of our simulation validation
- Potential for our approach to provide **practical guidance for future experiments**

## Summary of revision
In response to reviewer feedback, we strengthened the manuscript in the revision, addressing several important questions raised by the reviewers through additional analyses and discussions:
1. **New empirical analysis on real data using the Allen Brain Observatory dataset (Sec. 5, Fig. 7)**: We performed orientation decoding using the large scale public dataset and reported near-zero information gap (0.0024+/-0.064, p=0.63), which agrees with theoretical prediction of zero information gap under conventional task design of single-context. This empirical result highlights why existing datasets fail at distinguishing coding hypotheses.
2. Discussion on how our framework can **incorporate behavior measurements to account for biased priors/ model mismatch (A.5, Fig. 12)**: Through inferring biased internal priors from psychophysics curve and using the inferred priors for computing subject-specific information gap, our framework can be extended to sub-optimal observers by incorporating behavior data.
3. Discussion on how our framework **handles more nuanced mixed coding hypotheses (A.6, Fig. 13)**: In addition to pure likelihood and pure posterior coding hypotheses, our framework and optimized tasks can be extended to discriminate more nuanced, mixed coding hypotheses where information gap is predicted to be 0.
4. Analyses on firing rates, noise levels (A.7, Fig. 14), and factors affecting convergence (A.8, Fig. 15).
5. Analysis on thin-tailed distribution (A9, Fig. 16-17): demonstrating similar results to heavy-tailed context priors (near-zero posterior-coding information gap).
6. Example of simulated likelihood and posteriors (A.10, Fig. 18)
7. **Improved clarity and presentation**, including enhanced explanations of experimental paradigm (focus on well-trained steady state, contexts are explicitly cued) and information gap definition and derivation in the main text (Sec. 2).
8. Expanded related-work section as suggested (Sec. 1).

### Summary of contributions
Across reviews, we clarified the core contribution of our work bridging theory and experiment:
- The information gap: a new, analytically closed-form information-theoretic metric that quantifies the discriminability between likelihood and posterior coding hypotheses.
- A principled method for optimizing stimulus prior distributions to maximally separate the hypotheses—resolving task design tradeoffs that intuition or heuristic search cannot capture.
- Strong empirical validation, showing that decoder performance differences converge tightly to theoretical predictions across noise models, contrasts, and population sizes.

---

### Meta-Review · Area_Chair_hy6c · 2026-01-07

**Summary:**

The paper proposes a novel information-theoretical framework to address a fundamental debate in computational neuroscience: whether early sensory populations encode likelihood functions (e.g., Probabilistic Population Codes) or posterior distributions (e.g., Neural Sampling). The authors introduce the "Information Gap" metric, a quantitative measure of the expected cross-entropy difference between likelihood and posterior decoders. By maximizing this metric, the framework allows researchers to mathematically optimize experimental designs, specifically the separation and variance of context priors, to maximally differentiate these two coding hypotheses. The theoretical claims are validated via simulations using deep neural network decoders and Poisson neuron models.

The critical tension in the review process was between the mathematical soundness of the framework and its biological validity. The most substantial concern, raised by Reviewer 1gnw, is that biological sensory adaptation (e.g., gain control for metabolic efficiency) may confound the "steady-state" encoding assumptions required by the framework. If V1 responses change due to adaptation rather than Bayesian inference, the theoretical premises might not hold in vivo. Additionally, there was a collective concern regarding the lack of empirical validation; reviewers questioned whether the optimized designs would work in practice or if the framework remained purely speculative. The authors addressed the latter by analyzing the Allen Brain Observatory dataset, demonstrating that current single-context designs yield a near-zero information gap, effectively proving that existing data is insufficient to solve the coding debate.

I recommend acceptance. While the concerns regarding biological adaptation are valid, rejecting this work for lacking in vivo validation creates a "chicken and egg" problem: the field cannot validate these hypotheses without first having the experimental designs to do so. This paper provides that necessary "enabling technology." Regarding the adaptation critique (Reviewer 1gnw), recent literature (e.g., Tring et al.) actually supports the premise that V1 populations change responses based on stimulus statistics; the outstanding question is how to interpret those changes (efficiency vs. inference), which is exactly what this quantitative framework aims to disentangle. Furthermore, the authors’ analysis of the Allen Brain dataset compellingly demonstrates that current approaches are mathematically incapable of distinguishing between likelihood and posterior coding. On balance, the rigorous theoretical contribution provides a crucial blueprint for future experimental neuroscience.

**Reviewer Concerns:**

### Concerns addressed by the rebuttal:
* Reviewers UTZ6 and EZQq raised the need for empirical relevance. In particular, a major critique was the lack of real-world data application. In the revision, the authors analyzed the Allen Brain Observatory dataset. They empirically demonstrated that the information gap is effectively zero in this dataset (validating their theoretical prediction for single-context designs). This powerfully illustrates why existing data is insufficient and why their proposed multi-context design is necessary.
* Reviewer 1gnw had concerns about mixed coding models (e.g., Ganguli & Simoncelli). The authors addressed these concerns by showing these yield a distinct $\Delta_{info} \approx 0$ signature, allowing them to be analytically distinguished from pure hypotheses.
* Reviewers UTZ6 and JDGu both talked about sub-optimal observers. The authors added a procedure (Appendix A.5) to incorporate subject-specific biased priors inferred from psychometrics, making the framework robust to non-ideal behaviour.

### Outstanding concerns:
Reviewer 1gnw raised a substantive concern that biological sensory adaptation (e.g., gain changes for efficiency) might be conflated with posterior modulation, potentially violating the "steady-state" assumptions. While the authors argue that well-trained steady states allow for separation, this remains a significant experimental challenge.

**Reviewer Scores:**

I estimated scores based on the rebuttal with the authors.

* **Reviewer UTZ6. Original score: 4. Estimated score: 6**. This reviewer's primary request was for empirical testing on real neural data. The Allen Brain analysis directly fulfilled this. A score of 6 reflects that the main critique was addressed, though practical hurdles remain.
* **Reviewer 1gnw. Original score: 2. Estimated score: 4**. This reviewer is an expert on adaptation. While the authors clarified the theoretical stance, the fundamental disagreement regarding "steady-state" encoding versus adaptation likely persists. A score of 4 acknowledges the rebuttal effort while reflecting outstanding biological reservations.
* **Reviewer EZQq. Original score: 6. Estimated score: 8**. This reviewer was already positive regarding soundness. The clarifications on neural sampling and the inclusion of the requested empirical data justify a bump to "Good Paper."
* **Reviewer JDGu. Original score: 6. Estimated score: 6**. The reviewer liked the framework but questioned the practical "punchline." While the rebuttal clarified the framework's utility, sticking to the original score of 6 is a conservative estimate that acknowledges the lack of positive in-vivo validation.

---

### Decision · Program_Chairs · 2026-01-26

Accept (Poster)